# Hard Work Does Not Always Pay Off:
# On the Robustness of NAS to Data Poisoning

**Zachary Coalson**                                      *coalsonz@oregonstate.edu*
*School of Electrical Engineering and Computer Science*
*Oregon State University*
**Huazheng Wang**                               *huazheng.wang@oregonstate.edu*
*School of Electrical Engineering and Computer Science*
*Oregon State University*
**Qingyun Wu**                                              *qingyun.wu@psu.edu*
*College of Information Science and Technology*
*Penn State University*
**Sanghyun Hong**                             *sanghyun.hong@oregonstate.edu*
*School of Electrical Engineering and Computer Science*
*Oregon State University*

**Reviewed on OpenReview:** *https://openreview.net/forum?id=Uhayg3Ia9W*

## Abstract

We study the robustness of *data-centric methods* to find neural network architectures, known as neural architecture search (NAS), against data poisoning. To audit this robustness, we design a poisoning framework that enables the systematic evaluation of the ability of NAS to produce architectures under data corruption. Our framework examines four off-the-shelf NAS algorithms, representing different approaches to architecture discovery, against four data poisoning attacks, including one we tailor specifically for NAS. In our evaluation with the CIFAR-10 and CIFAR-100 benchmarks, we show that NAS is *seemingly* robust to data poisoning, showing marginal accuracy drops even under large poisoning budgets. However, we demonstrate that when considering NAS algorithms designed to achieve a few percentage points of accuracy gain, this expected improvement can be substantially diminished under data poisoning. We also show that the reduction varies across NAS algorithms and analyze the factors contributing to their robustness. Our findings are: (1) Training-based NAS algorithms are the least robust due to their reliance on data. (2) Training-free NAS approaches are the most robust but produce architectures that perform similarly to random selections from the search space. (3) NAS algorithms can produce architectures with improved accuracy, even when using out-of-distribution data like MNIST. We lastly discuss potential countermeasures.

## 1 Introduction

Recent years have seen success stories of a *data-centric* approach to designing neural network architectures, namely neural architecture search (NAS) (Liu et al., 2018b; Real et al., 2019; Liu et al., 2018a; Zoph & Le, 2017; Zoph et al., 2018). Given a dataset and a predefined search space of architectural choices, NAS algorithms iteratively explore architectures that optimize specific objectives. A common objective is to maximize validation accuracy: during search, NAS selects architectures that perform the best on the data. This approach has shifted the paradigm from manual engineering to automated neural network design, facilitating the improvement of seminal architectures such as Transformers (So et al., 2019).

Research has warned that approaches to "learning" system internals from external data may introduce a critical risk—*data poisoning.* A data poisoning adversary compromises a subset of the training data with the objective of inducing malicious outcomes. Most existing work on poisoning attacks focuses on adversaries who *manipulate* the training process of neural networks and induce multitudes of adversarial effects to trained

models, such as performance degradation (Steinhardt et al., 2017; Lu et al., 2022; 2023; Zhao & Lao, 2022), backdoors (Geiping et al., 2021; Huang et al., 2020; Shafahi et al., 2018; Aghakhani et al., 2021), or the leakage of sensitive information (Wen et al., 2024; Chen et al., 2022). However, to the best of our knowledge, it remains an open question whether an adversary can, by *only* compromising the data—without altering the search algorithm—negate the benefits of NAS and cause it to produce "sub-optimal" architectures.

In this work, we address this knowledge gap by studying the following research question: *How vulnerable are NAS algorithms to data poisoning attacks?* We particularly focus on the performance improvements achieved by architectures produced by NAS and assess how data poisoning can undermine these gains. Poisoning adversaries can introduce adversarial distributional shifts, such that when the compromised data is used for architecture search, the victim NAS algorithm generates architectures that deviate significantly from optimal, leading to, for instance, degraded performance. This threat is particularly alarming because NAS is purely empirical, making it challenging to identify whether generated architectures are truly sub-optimal.

## 1.1 Contributions

We *first* present a framework for systematically studying the robustness of NAS algorithms against data poisoning. Designed for auditing, the framework operates under a white-box assumption, in which the adversary knows the NAS algorithms. But, certain *black-box* poisoning attacks do not require such knowledge. Our framework includes four attacks, one of which is specifically tailored for NAS. We also develop a testing methodology and a metric to quantify the performance gains of architectures generated by NAS.

*Second*, we comprehensively evaluate NAS algorithms using our framework. To conduct a systematic comparison, we carefully choose four recent NAS algorithms that share similar search spaces but operate using different paradigms: training-based, training-free, and hybrid. We run each attack ten times, exceeding the standard practices established in prior work (Liu et al., 2019). We make the following intriguing findings:

(1) The NAS algorithms we examine are *seemingly* robust to data poisoning attacks, when evaluated using the standard metric, accuracy. But, when assessed with our proposed metric, their performance gains are significantly degraded—by as much as 156%.

(2) Training-based NAS is the most vulnerable to poisoning, whereas training-free methods show the most resilience. However, when run on clean data, training-based methods produce architectures with the highest accuracy, followed by hybrid approaches, while training-free methods achieve the lowest accuracy, suggesting a potential utility-robustness trade-off.

(3) Our in-depth analysis raises open questions. Training-free algorithms show insensitivity to the training distribution and generate architectures that perform comparably to those randomly selected from the search space, questioning their ability to find high-quality, dataset-specific architectures.

(4) The NAS algorithms we examine, even when run on out-of-distribution data—such as using MNIST for discovering CIFAR-10 architectures—still produce architectures that perform well, and in some cases, even outperform those found using CIFAR-10 itself.

We *lastly* review existing defenses against poisoning attacks, with a focus on their applicability to NAS. We evaluate three defenses and demonstrate that they are ineffective in mitigating the impact of data poisoning.

## 2 Preliminaries

**Neural architecture search (NAS)** is an automated technique for designing neural network architectures. Given a dataset, search space, and objective, NAS algorithms iteratively sample architectures from the search space and select the best-performing candidates. The search space defines the possible architectural choices, including layer types, configurations, and connections. The sampling process is designed to optimize the given objective, typically validation performance (Liu et al., 2019; Pham et al., 2018; Zoph & Le, 2017). But recent studies incorporate additional objectives, such as latency and computational efficiency (Cai et al., 2019; Tan et al., 2019; Lu et al., 2019; 2020).

**Differentiable NAS** enhances efficiency by formulating architecture search as a continuous optimization problem (Liu et al., 2019; Cai et al., 2019; Xu et al., 2020). Instead of sampling discrete candidates, these algorithms train a *supernet* that encodes all possible architectures. The supernet includes *architectural parameters* that weight different operations in the forward pass. These operations are commonly fused as:

$$\bar{o}^{(i,j)}(x) = \sum_{o \in \mathcal{O}} \frac{\exp\left(\alpha_o^{(i,j)}\right)}{\sum_{o' \in \mathcal{O}} \exp\left(\alpha_{o'}^{(i,j)}\right)} o(x),$$

where $\mathcal{O}$ is the set of candidate operations, and $\alpha_o^{(i,j)}$ is a parameter that weights operation $o$ between layers $i$ and $j$. This formulation enables the simultaneous bi-level optimization of architectural parameters and supernet weights. A final *discretized* architecture is derived by selecting the highest-weighted operations.

**Training-free and hybrid NAS** are more recent approaches that forgo architecture training and instead assess candidates with efficient-to-compute *training-free metrics* (He et al., 2024; Chen et al., 2021; Shu et al., 2022b;a). These metrics estimate architecture quality based on expressivity (Xiong et al., 2020), trainability (Jacot et al., 2018), and other data-agnostic properties and can be computed orders of magnitude faster than full model training. *Training-free* NAS selects architectures solely based on these metrics, bypassing model training. *Hybrid* NAS balances efficiency and performance by incorporating limited training—e.g., to inform Bayesian optimization-guided search (Shu et al., 2022b; He et al., 2024; Shen et al., 2023).

**Data poisoning** is a training-time attack where the attacker injects carefully crafted poisoned samples into the victim's training data to induce the models trained on the compromised data to behave maliciously. A typical objective of poisoning attacks is indiscriminate performance degradation (Steinhardt et al., 2017; Zhao & Lao, 2022; He et al., 2023; Lu et al., 2023), while targeted attacks cause misclassification of specific test samples (Shafahi et al., 2018; Aghakhani et al., 2021; Huang et al., 2020; Geiping et al., 2021), increase privacy leakage of sensitive training data (Wen et al., 2024; Chen et al., 2022), or teach models to misclassify inputs with a learned trigger pattern (known as backdooring) (Gu et al., 2019; Liu et al., 2018c). Our work studies indiscriminate attacks in a new setting: we deviate NAS algorithms from finding optimal architectures.

## 3 The Auditing Framework

### 3.1 Threat Model

We consider a data-poisoning adversary that exploits the inherent "data-centric" vulnerability of NAS. The attacker aims to degrade the performance of architectures produced by the victim's NAS algorithm when it is run on tampered data. By manipulating the data used in the search process, the adversary can mislead NAS into selecting suboptimal architectures. This is particularly problematic because there is no *oracle* that can determine the best or worst architectures NAS can discover before running the search. As a result, the victim must trust that the identified architecture is optimal, even if it has been manipulated.

**Capabilities.** We assume that the attacker has the capability to compromise the dataset used for running NAS algorithms. Recent studies demonstrate that large-scale data poisoning is practical (Carlini et al., 2024), e.g., an attacker can host malicious images on publicly accessible websites, such as GitHub Pages, to increase the likelihood that the victim will collect and use these poisoned samples. The scale makes it challenging to manually filter out suspicious samples. We denote the attacker's poisoning budget as $p$. Prior studies typically consider $p \leq 1\%$, which we adopt as our *practical* budget. We further explore *stress-testing* scenarios where the adversary compromises up to 50% of the dataset to reveal each attack's worst-case outcomes.

**Knowledge.** Because we focus on auditing, we assume a white-box adversary with knowledge of the victim's NAS algorithm. However, we emphasize that only one of the four poisoning attacks we evaluate requires this assumption, while the remaining three can operate in a black-box manner.

### 3.2 (Our) Data Poisoning Attacks

We conduct our evaluation on image classification benchmarks, where existing poisoning attacks against models trained for this task are broadly categorized into two types: *dirty-label* attacks, which manipulate

the labels of training samples, and *clean-label* attacks, which subtly modify images while preserving labels. A naive attacker may not employ a sophisticated strategy but instead *randomly* manipulate either labels or images. In contrast, an advanced adversary leverages carefully-designed poison-crafting algorithms to maximize the impact of their poisoning samples. We consider these axes comprehensively and include the following four poisoning attacks in our framework:

**Dirty-label poisoning attacks.** The first attack we consider is *random label-flipping* (RLF), which flips training data labels to randomly chosen alternatives. RLF serves as a measure of the impact of label noise on NAS performance (Northcutt et al., 2021).

In contrast, a sophisticated adversary may flip labels in a manner that maximizes performance degradation. To test against this adversary, we also develop *confidence-based label-flipping* (CLF). CLF attacks select labels to flip based on the confidence of a pre-trained (surrogate) neural network. We first rank all the training samples based on the surrogate model's prediction probabilities—i.e., the logit value of each sample's most likely class. Next, we flip the labels to the least likely classes. By compromising samples that a surrogate model classifies with high confidence, CLF increases task complexity, making it more difficult for NAS to effectively "learn" optimal architectures compared to RLF. We show the detailed algorithm in Algorithm 1.

---

**Algorithm 1** Confidence-Based Label-Flipping

**Require:** Surrogate model $F(\cdot)$, training images and labels $(x_i, y_i)_{i=1}^n$, poisoning budget $p$.
 1: Initialize empty array for logits: $\mathcal{Z} \leftarrow \{\}$
 2: **for** $i = 1, 2, \ldots, n$ **do**
 3:     $z_i \leftarrow F(x_i)$
 4:     $\mathcal{Z}$.append($(z_i, x_i)$)                                   // store logits of $i^{th}$ training example
 5: **end for**
 6: Sort $\mathcal{Z}$ by $\max(z_i)$ for all $(z_i, x_i)$ in decreasing order // sort by maximum logit to find the most confident samples
 7: Initialize array for poisons: $\mathcal{P} \leftarrow \{\}$
 8: **for** $i = 1, 2, \ldots, np$ **do**
 9:     $(z_i', x_i') \leftarrow \mathcal{Z}_i$
10:     $y_i' = \mathrm{argmin}(z_i')$                                   // set new label to least confident class
11:     $\mathcal{P}$.append($(x_i', y_i')$)
12: **end for**
13: **return** poisoned samples $\mathcal{P}$

---

**Clean-label poisoning attacks.** We first consider *Gaussian noise*, which represents the weakest form of a clean-label attack. It adds Gaussian random noise $\sim \mathcal{N}(0, \sigma^2 \mathbb{I})$ to datasets. While not typically classified as a data poisoning attack, it has been extensively studied as an image-level corruption (Hendrycks & Dietterich, 2019; Rusak et al., 2020). To ensure meaningful perturbations without overly degrading image quality, we set $\sigma$ to 16 and bound the noise using an $\ell_\infty$-norm of 16-pixels.

We also employ *gradient canceling* (GC) (Lu et al., 2023), the current state-of-the-art, clean-label, indiscriminate attack. It first utilizes the GradPC framework (Sun et al., 2020) to generate a set of target parameters (weights and biases) that are close to the model's original parameters, yet lead to degraded performance. GC then crafts poisoning samples that, when injected into the training data, manipulate the training dynamics. They induce gradient updates that counteract those from clean data, effectively forcing the model to converge to the target parameters generated by GradPC.

Moreover, we adapt the existing GC attacks for NAS, introducing *NAS-specific GC*, which varies on the victim's NAS algorithm. For training-free and hybrid NAS, instead of crafting poisoning samples based on predefined target model parameters, we generate them using the parameters of the final architectures trained on clean data. For training-based NAS algorithms, we adapt GC to target the architecture parameters from which gradient-based poisoning can be derived. This ensures that the poisoning process directly influences the architecture selection process, rather than targeting a fixed-trained model. Here, we assume the attacker knows the victim's NAS algorithm and its selection process.

---

**Algorithm 2** Gradient Canceling Lu et al. (2023)

---

**Require:** Victim model $\{F(\cdot), \theta\}$, training images and labels $(x_i, y_i)_{i=1}^n$, poisoning budget $p$, perturbation bound $\varepsilon$, number of optimization steps $S$.

1: Randomly sample a subset $\mathcal{P} \subseteq \{x_i, y_i\}_{i=1}^n$ of size $np$ as poisons
2: Initialize perturbation mask: $\Delta \in \mathbb{R}^{np \times C}$     $// C$ is the size of one image
3: Compute $g_{\text{clean}} \leftarrow \frac{1}{n} \sum_{i=1}^n \nabla_\theta \mathcal{L}_{\text{xe}}(F(x_i), y_i)$
4: **for** $t = 1, 2, \ldots S$ **do**
5:     Apply poison mask: $\mathcal{P}' \leftarrow (x_i + \Delta_i)_{x_i \in \mathcal{P}}$
6:     Compute $g_{\text{adv}} \leftarrow \frac{1}{|\mathcal{P}'|} \sum_{(x_i, y_i) \in \mathcal{P}'} \nabla_\theta \mathcal{L}_{\text{xe}}(F(x_i), y_i)$
7:     Compute loss: $L \leftarrow \frac{1}{2} \|(1-p) \cdot g_{\text{clean}} + p \cdot g_{\text{adv}}\|_2^2$     $//$ "cancels" clean gradients with poisons
8:     Update $\Delta$ with a step of Adam and project onto $\|\Delta\|_\infty \leq \varepsilon$
9: **end for**
10: **return** poisoned samples $(x_i + \Delta_i, y_i)_{x_i \in \mathcal{P}}$

---

We present the final attack in Algorithm 2. We adopt the official GC implementation (Lu et al., 2023) with a few minor modifications. The original work uses SGD with momentum for optimization, but we employ Adam (Kingma & Ba, 2017) as we find it consistently produces lower-loss poisons. It also does not consider the standard perturbation bounds in the clean-label literature (Huang et al., 2020; Geiping et al., 2021; Aghakhani et al., 2021). To achieve consistency with prior work, we introduce an $\ell_\infty$-norm bound moderated by the hyperparameter $\varepsilon$. We did not observe substantial differences in attack success when using $\varepsilon = 16$.

For TE-NAS, RoBoT, and NSGANetV2, $F(\cdot)$ is a fully trained architecture from the search space, and $\theta$ its weights and biases. To craft poisons on P-DARTS, $F(\cdot)$ is set to a converged supernet and $\theta$ its architectural parameters controlling the weight of operations. In both cases, we first perturb the target parameters $\theta$ using GradPC (Sun et al., 2020), which performs bounded parameter updates that *increase* the cross-entropy loss to decrease performance. We refer readers to the original work (Lu et al., 2023) for more details.

### 3.3 Testing Methodology and Metric

**Methodology.** Our primary objective is to evaluate the impact of data poisoning on architectures produced by NAS algorithms. To achieve this, we carefully design our testing methodology to isolate the impact of poisoning samples on the architecture selection process while ensuring that model training remains unaffected. In all our experiments, we run NAS algorithms on the tampered data to select architectures, but we train the generated architectures from scratch using *clean* data. This approach makes sure that any observed degradation in performance is attributable solely to the influence of poisons on NAS, rather than on the training process itself—i.e., model parameters are not affected by poisons.

**Metrics.** The standard metric for assessing the impact of poisoning attacks is classification accuracy (or accuracy). A straightforward approach to measuring attack effectiveness is to evaluate the accuracy of architectures produced by NAS algorithms after being trained from scratch on clean data. However, accuracy alone is not a sufficient metric to fully capture the impact of data poisoning on architectures. NAS algorithms are not designed to transform *poor* architectures into *good* architectures; rather, they aim to refine good architectures into great ones. Even without NAS, existing architectures (or architectures randomly sampled from the search space) already achieve over 95% accuracy on CIFAR-10, and the improvements reported in prior work are typically a few percentage points, e.g., from 95% to 97%. To better quantify the impact of poisoning attacks on the *improvement* NAS provides, we design an additional metric ΔImprovement (ΔImp.): the *percentage point change* in accuracy between the architecture produced by a NAS algorithm running on tampered data and an architecture randomly sampled from the same search space. ΔImp. better captures how poisoning attacks degrade the intended performance gains of NAS.

### 3.4 Selection of NAS Algorithms

We consider NAS algorithms based on the following criteria: (1) Comprehensiveness—we ensure the inclusion of diverse NAS approaches: training-based, training-free, and hybrid algorithms; (2) Recency and

representativeness—we prioritize NAS algorithms that have been recently introduced and are gaining popularity; and (3) Computational efficiency—because running NAS algorithms can take from a few to hundreds of GPU hours, we select those that can be efficiently run within a few hours on a single GPU.

**P-DARTS** (Chen et al., 2019) is a *training-based* algorithm that builds upon the seminal work DARTS (Liu et al., 2019). It employs a differentiable supernet and optimizes *architectural parameters*—which determine the importance of different operations—via gradient descent. The key advance from DARTS is a *progressive* search strategy, which gradually increases the supernet's layer count, allowing the search to occur in a space more representative of final architectures. Both P-DARTS and DARTS are popular differentiable NAS, but DARTS (Liu et al., 2019) suffers from computational complexity and training instability (Chen et al., 2019; Xu et al., 2020; Chu et al., 2021). P-DARTS addresses these problems and runs faster than DARTS.

**NSGANetV2** (Lu et al., 2020) is a training-based method that deploys an *evolutionary algorithm* (EA) to generate candidate architectures. It integrates two surrogate models: an upper-level surrogate that predicts architecture performance to guide the NSGA-II (Deb et al., 2002) evolutionary search, and a lower-level weight-sharing supernet for efficient training and evaluation of candidate architectures. NSGANetV2 is also *multi-objective* and optimizes multiple networks simultaneously with different performance–size trade-offs. It is substantially more efficient than initial EA-based approaches Lu et al. (2019); Real et al. (2019) while offering improved performance, making it tractable to consider in our study.

**TE-NAS** (Chen et al., 2021) is a *training-free* algorithm that evaluates a network's performance based on trainability and expressivity. Trainability is measured using the condition number of the neural tangent kernel (Jacot et al., 2018), while expressivity is quantified by the number of linear regions in the activation space. These metrics guide an iterative pruning process on a NAS supernet until a final, discretized architecture is produced. Recent studies (He et al., 2024; Shu et al., 2022b) find TE-NAS to be one of the most effective training-free algorithms, which, along with its popularity, makes it a strong candidate for our study.

**RoBoT** (He et al., 2024) is a *hybrid* NAS that combines training-free metrics with Bayesian optimization (BO). It first samples architectures from the search space and computes multiple training-free metrics. BO then optimizes a weighted combination of these metrics based on validation accuracy (measured after a few training epochs) as the objective function. The architecture with the highest weighted score is chosen. We choose RoBoT for our study as it is the most recent and effective hybrid NAS. It also employs a BO framework similar to prior works (Shen et al., 2023; Shu et al., 2022b), making it representative.

## 4 Empirical Evaluation

Now we utilize our framework to audit the robustness of NAS against data poisoning.

### 4.1 Experimental Setup

**Hardware and software.** We use Python[1] with PyTorch[2] for all experiments, with version varying depending on the NAS algorithm. We run experiments on a system with a 48-core Intel Xeon Processor, 768GB of memory, and 8 NVIDIA A40 GPUs.

**NAS algorithms.** We use the official PyTorch implementation of all NAS algorithms (Chen et al., 2021; He et al., 2024; Chen et al., 2019; Lu et al., 2020) introduced in §3.4. We use the same hyperparameters as in the original works and add functionality for inserting poisons before searching. To obtain a fully trained network, we run the NAS search algorithm on the poisoned training data, save the final architecture, and retrain the architecture from scratch for 600 epochs on clean data. We retrain all architectures using the P-DARTS (Chen et al., 2019) training implementation with the default hyperparameters.

**NAS search space.** We adopt the popular DARTS search space (Liu et al., 2019), which is defined by three factors: the number of internal nodes, the incorporation of prior states, and the set of candidate operations. These factors are used to build a *cell*, which is stacked to build the final architecture after searching. The

---

[1]Python: https://www.python.org
[2]PyTorch: https://pytorch.org/

candidate operations are: {none, 3x3 max pooling, 3x3 average pooling, skip connection, 3x3 separable convolution, 5x5 separable convolution, 3x3 dilated convolution, 5x5 dilated convolution}, where *none* denotes the absence of a connection between two nodes. Following prior work, we search over architectures with four internal nodes, each of which incorporates the previous two states (i.e., candidate operations from those states are applied to the internal nodes). The original work estimates that this search space contains $\sim 10^{18}$ possible architectures, without accounting for graph isomorphisms.

**Poison crafting.** We run our evaluation with CIFAR-10 and CIFAR-100 (Krizhevsky, 2009). To craft poisons, we randomly select $\{1, 10, 50\}\%$ of the training data and run the poisoning attacks from §3.2. In our CLF attack, we use a ResNet-18 (He et al., 2016) model trained on each dataset as the surrogate classifier. The target parameters for GC depend on the NAS algorithm. For TE-NAS, RoBoT, and NSGANetV2, we use the weights and biases of the final architectures, while for P-DARTS, we use the architectural parameters of the supernet. For both, we set the highest-accuracy networks on clean data (across 10 trials) as the target and craft poisons for 250 steps. Following the standard setting in prior work (Huang et al., 2020; Geiping et al., 2021; Aghakhani et al., 2021), we apply $\ell_\infty$-norm bounds of $\varepsilon = 16$ to all clean-label perturbations. Appendix C shows examples of the crafted poisons and their clean counterparts from CIFAR-10.

Table 1: **Robustness of NAS to data poisoning on CIFAR-10 and CIFAR-100.** We report the accuracy (**Acc.**) and percentage-point change from random sampling (**$\Delta$Imp.**). $p$ is the poisoning budget. Each cell contains the mean and standard deviation. We **underline and bold** statistically significant results.

| Poisoning Attack | $p$ | CIFAR-10 | | | | | | CIFAR-100 | | | | | |
|---|---|---|---|---|---|---|---|---|---|---|---|---|---|
| | | P-DARTS | | RoBoT | | TE-NAS | | P-DARTS | | RoBoT | | TE-NAS | |
| | | Acc. | $\Delta$Imp. | Acc. | $\Delta$Imp. | Acc. | $\Delta$Imp. | Acc. | $\Delta$Imp. | Acc. | $\Delta$Imp. | Acc. | $\Delta$Imp. |
| None | – | $97.32 \pm 0.07$ | 0.58 | $96.99 \pm 0.20$ | 0.25 | $96.80 \pm 0.40$ | 0.07 | $82.58 \pm 0.62$ | 1.88 | $81.68 \pm 0.41$ | 0.98 | $81.40 \pm 0.73$ | 0.71 |
| Gaussian Noise | 1% | $97.33 \pm 0.12$ | 0.60 | $97.10 \pm 0.10$ | 0.36 | $96.91 \pm 0.29$ | 0.17 | $82.89 \pm 0.26$ | 2.19 | $81.65 \pm 0.62$ | 0.96 | $82.02 \pm 0.56$ | 1.33 |
| | 10% | **$\underline{97.17}$** $\pm 0.13$ | **$\underline{0.44}$** | $97.08 \pm 0.12$ | 0.35 | $96.67 \pm 0.46$ | -0.07 | $82.38 \pm 0.35$ | 1.69 | $81.51 \pm 0.75$ | 0.81 | $81.69 \pm 0.37$ | 0.99 |
| | 50% | $97.26 \pm 0.11$ | 0.53 | $97.03 \pm 0.04$ | 0.30 | $96.71 \pm 0.31$ | -0.02 | $82.40 \pm 0.45$ | 1.70 | $81.25 \pm 0.45$ | 0.56 | $82.08 \pm 0.69$ | 1.39 |
| GC | 1% | **$\underline{97.19}$** $\pm 0.15$ | **$\underline{0.46}$** | $97.01 \pm 0.15$ | 0.28 | $96.85 \pm 0.28$ | 0.12 | $82.68 \pm 0.40$ | 1.99 | $81.82 \pm 0.46$ | 1.12 | $81.83 \pm 0.47$ | 1.14 |
| | 10% | $97.21 \pm 0.18$ | 0.48 | $97.20 \pm 0.18$ | 0.47 | $96.86 \pm 0.23$ | 0.13 | $82.39 \pm 0.53$ | 1.70 | $81.69 \pm 0.25$ | 0.99 | $81.61 \pm 0.64$ | 0.91 |
| | 50% | **$\underline{97.15}$** $\pm 0.15$ | **$\underline{0.42}$** | $97.13 \pm 0.08$ | 0.40 | $96.91 \pm 0.31$ | 0.18 | $82.32 \pm 0.45$ | 1.62 | $81.94 \pm 0.83$ | 1.25 | $81.47 \pm 0.62$ | 0.77 |
| RLF | 1% | $97.23 \pm 0.23$ | 0.50 | $97.13 \pm 0.23$ | 0.39 | – | – | $82.73 \pm 0.39$ | 2.03 | $81.63 \pm 0.26$ | 0.93 | – | – |
| | 10% | $97.22 \pm 0.19$ | 0.49 | $96.95 \pm 0.13$ | 0.21 | – | – | $82.18 \pm 0.42$ | 1.48 | $81.51 \pm 0.24$ | 0.81 | – | – |
| | 50% | **$\underline{96.77}$** $\pm 0.38$ | **$\underline{0.04}$** | $97.04 \pm 0.13$ | 0.30 | – | – | **$\underline{79.64}$** $\pm 1.83$ | **$\underline{-1.05}$** | $81.04 \pm 0.71$ | 0.34 | – | – |
| CLF | 1% | **$\underline{97.11}$** $\pm 0.20$ | **$\underline{0.38}$** | $97.03 \pm 0.31$ | 0.30 | – | – | $82.60 \pm 0.50$ | 1.91 | $81.70 \pm 0.45$ | 1.00 | – | – |
| | 10% | $97.27 \pm 0.11$ | 0.54 | $96.88 \pm 0.06$ | 0.15 | – | – | $82.66 \pm 0.39$ | 1.96 | $81.41 \pm 0.48$ | 0.71 | – | – |
| | 50% | **$\underline{97.04}$** $\pm 0.29$ | **$\underline{0.31}$** | $97.15 \pm 0.04$ | 0.42 | – | – | **$\underline{81.59}$** $\pm 0.88$ | **$\underline{0.89}$** | $81.89 \pm 0.28$ | 1.19 | – | – |

## 4.2 Effectiveness of Data Poisoning Attacks

We first sample 10 random architectures from the DARTS search space and train them from scratch on CIFAR-10 and CIFAR-100, achieving average test-time accuracies of 96.73% and 80.70%, respectively. These serve as our *random sampling baseline* for computing the $\Delta$Imp. metric. We then conduct our data poisoning attacks on each NAS algorithm across 10 trials and report the accuracy (**Acc.**) and the percentage-point difference in accuracy relative to random sampling (**$\Delta$Imp.**). A poisoning attack is considered to be *effective* if it reduces $\Delta$Imp. below the level achieved under clean data. To evaluate whether this effect is statistically significant, we conduct one-sided Welch's t-tests comparing the mean accuracies of poisoned architectures to their clean counterparts (**No Attack**) at a significance level of $\alpha = 0.05$. The null hypothesis states that poisoning does not reduce accuracy, and we reject it when $p \leq 0.05$. For each model, we control the false discovery rate (FDR) at $\alpha = 0.05$ using the Benjamini–Hochberg (BH) procedure applied to all p-values. We denote statistically significant results by **underlining and bolding** them. Results for P-DARTS, TE-NAS, and RoBoT are shown in Table 1. As NSGANetV2 is substantially more compute-intensive—requiring $\sim 5\times$ more search time than P-DARTS—we run a subset of our evaluation and report the results in Appendix A.

**NAS is *seemingly* robust to data poisoning.** Our results show that NAS appears to be robust to data poisoning: in the instances where Acc. is reduced, it drops by only 0.01–2.93% points. The state-of-the-art indiscriminate poisoning attack reduces Acc. by $\sim$15% points on standard neural networks (Lu et al., 2023), suggesting that NAS is 5–1500$\times$ more resilient. However, comparing this degradation to the *benefit* NAS provides over random sampling reveals that several attacks significantly reduce the expected improvement.

Specifically, the best-performing algorithm, P-DARTS, offers only a 0.58% and 1.88% point improvement over random sampling for CIFAR-10 and CIFAR-100. Across both datasets, the RLF and CLF attacks induce statistically significant reductions in $\Delta$Imp., lowering it to 0.38 at the practical budget of $p = 1\%$ and to -1.05 to 0.89 under stress-testing scenarios ($p \geq 10\%$). These drops correspond to a 34–156% reduction in the benefit provided by NAS. While poisoning causes small absolute drops in accuracy, these reductions can be substantial relative to the gains these algorithms typically provide.

**Label-flipping attacks are effective against training-based NAS.** The impact of label-flipping attacks varies widely across NAS algorithms. For P-DARTS, random label-flipping (RLF) does not significantly change $\Delta$Imp. for $p \leq 10\%$. However, at $p = 50\%$, $\Delta$Imp. drops to 0.04 on CIFAR-10 and -1.05 on CIFAR-100—statistically significant reductions that either eliminate P-DARTS's benefit or make it perform worse than random sampling. Confidence-based label-flipping (CLF) appears slightly more effective across all poisoning budgets, achieving statistically significant drops in $\Delta$Imp. to 0.38 when $p = 1\%$ and 0.31–0.89 when $p = 50\%$. However, it does not degrade performance to random sampling; at high poisoning budgets, random label noise appears more detrimental than targeted flips. RoBoT shows greater robustness, with no reductions that reach statistical significance. Since TE-NAS is inherently robust to label-flipping, training-based algorithms like P-DARTS are especially vulnerable to this attack vector.

**Clean-label attacks are not effective.** We find that Gaussian noise does not lower $\Delta$Imp. at the practical poisoning budget of $p = 1\%$. This is expected, as Gaussian noise tends to improve the generalization and robustness of neural networks when added to training data (Bishop, 1995; Franceschi et al., 2018; Rosenfeld et al., 2020). However, excessive noise can degrade NAS performance: at $p = 10\%$, $\Delta$Imp. drops to 0.44 for P-DARTS on CIFAR-10, a statistically significant result. Surprisingly, Gradient canceling (GC), the *targeted* clean-label attack, is largely ineffective. The changes in $\Delta$Imp. are small across most algorithms, and in several cases we observe slight increases. Our adapted P-DARTS implementation, which directly targets the architectural parameters, performs best—achieving statistically significant reductions in $\Delta$Imp. to 0.42–0.46 on CIFAR-10 when $p = 1\%$ and 50%—though still underperforming label-flipping. A possible explanation for this result is the usage of data augmentations (e.g., random crops and horizontal flips) used to improve generalization (Perez & Wang, 2017). All NAS algorithms in this study employ data augmentations, which reduce the effectiveness of clean-label attacks (Geiping et al., 2021; Schwarzschild et al., 2021; Borgnia et al., 2021). We quantify the impact of data augmentations on NAS-specific GC in Appendix B, and discuss additional limitations and opportunities for advancing NAS-specific data poisoning in §6.2.

**The utility-robustness tradeoff.** NAS robustness to data poisoning correlates with reliance on training data. P-DARTS, the most training-intensive algorithm, experiences the largest reductions in $\Delta$Imp. (up to 2.93) and is the only algorithm for which poisoning produced a statistically significant effect. It is also the only algorithm where data poisoning *completely* negated the expected improvement over random sampling (RLF with $p = 50\%$). In contrast, RoBoT and TE-NAS are far more robust: both exhibit only minor, non-significant $\Delta$Imp. reductions and, in most cases, maintain performance comparable to the clean setting. Interestingly, when we examine the $\Delta$Imp. of these algorithms on clean data, we see that higher data reliance correlates with greater performance. P-DARTS is best with a $\Delta$Imp. of 0.58, compared to 0.25 for RoBoT and 0.07 for TE-NAS, which barely outperforms random sampling. These findings suggest a *utility-robustness tradeoff*: higher data reliance improves performance but increases susceptibility to data poisoning.

## 5 Understanding the Robustness

This section analyzes the factors that influence the robustness between different NAS algorithms. We first examine how poisoning affects the architectures produced by NAS algorithms and what makes an architecture "sub-optimal". Next we analyze the robustness of the training-free metrics used in hybrid and training-free NAS. We finally test the ability of NAS to generalize to out-of-distribution data.

### 5.1 Impact of Data Poisoning on NAS Architectures

To explore the extent to which data poisoning affects NAS and why certain poisoned architectures perform worse, we conduct a qualitative analysis using architectures found in §4.2 on CIFAR-10. In Figure 1, we

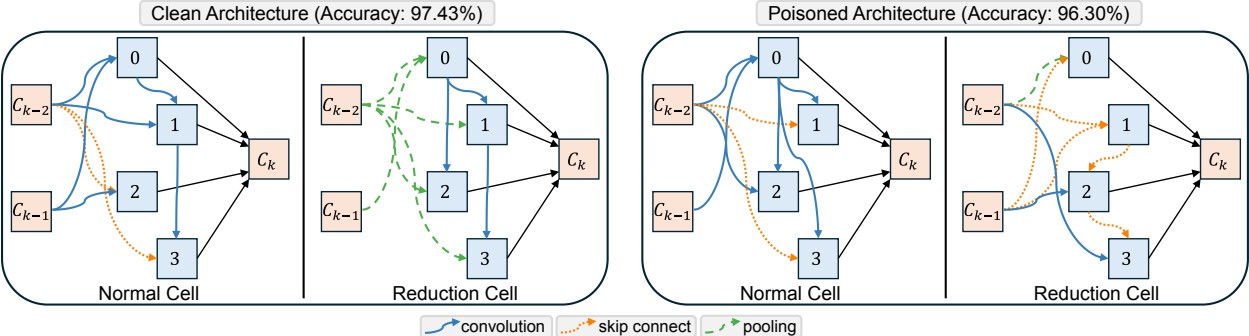

Figure 1: **Comparison of clean and poisoned architectures.** We visualize the best-performing architecture found using clean data (left) and the worst-performing architecture found using poisoned data (right) with the P-DARTS algorithm. $C_k$ is the output of cell $k$, and the numbered squares are the internal nodes (layers). The worst-performing architecture was found using RLF poisons with a budget of 50%.

visualize the best architecture found by P-DARTS on clean data (based on Acc.) and compare it to the worst architecture found on poisoned data ( RLF with $p = 50\%$). We omit visualizations for TE-NAS and RoBoT because none of the poisoning attacks resulted in a statistically significant reduction in Acc.

The clean architecture, with an Acc. of 97.43%, consists primarily of convolutions in the normal cell and pooling operations in the reduction cell. This is expected, as the normal cell is designed to extract features, followed by the reduction cell reducing their dimension. It also includes just two skip-connections, as they facilitate better learning in deeper networks (He et al., 2016) but can degrade performance if used in excess (Chen et al., 2019). The poisoned architecture has a similar normal cell with the same number of convolutions and skip-connections. However, the ∼1% point lower Acc. can likely be attributed to its reduction cell, which has only one pooling operation and, instead, mostly skip-connections. This architecture was produced on heavily label-flipped data, so the introduction of contradictions between features and their associated classes may have caused P-DARTS to select more skip-connections as they have less propensity to introduce error. While this result highlights just one instance of a sub-optimal architecture induced by data poisoning, it shows that these attacks are capable of substantially altering the architecture selection process.

### 5.2 Training-Free Metrics are Robust

Table 2: **Sensitivity of training-free metrics.** For each data poisoning attack, we report the average change in each training-free metric across 100 randomly sampled architectures. We use $p$=50% for all attacks.

| Poisoning Attack | TE-NAS | | RoBoT | | | |
|---|---|---|---|---|---|---|
| | $\kappa_{\mathrm{NTK}}$ | $\mathcal{R}$ | $grad\_norm$ | $snip$ | $grasp$ | $fisher$ |
| **Gaussian Noise** | $2.18 \pm 51.11\%$ | $2.27 \pm 9.03\%$ | $-0.69 \pm 0.76\%$ | $-0.54 \pm 0.67\%$ | $1.08 \pm 10.59\%$ | $2.57 \pm 1.31\%$ |
| **GC** | $65.42 \pm 679.35\%$ | $3.38 \pm 8.95\%$ | $-0.47 \pm 0.67\%$ | $-0.36 \pm 0.60\%$ | $0.52 \pm 5.03\%$ | $2.22 \pm 1.21\%$ |
| **RLF** | $0.00 \pm 0.00\%$ | $0.13 \pm 9.45\%$ | $-5.62 \pm 2.10\%$ | $-5.85 \pm 2.02\%$ | $-7.49 \pm 54.39\%$ | $0.32 \pm 1.45\%$ |
| **CLF** | $0.00 \pm 0.00\%$ | $0.89 \pm 9.04\%$ | $-5.04 \pm 4.27\%$ | $-6.66 \pm 3.52\%$ | $-11.14 \pm 119.65\%$ | $-1.87 \pm 4.37\%$ |

We hypothesize that the resilience of training-free and hybrid NAS to data poisoning results from the *insensitivity* of training-free metrics to distribution shifts. If the values of these metrics are not sufficiently altered under poisoning, the algorithms that deploy them will not be affected. To test this hypothesis, we evaluate the sensitivity of all training-free metrics[3] used by TE-NAS and RoBoT to our data poisoning attacks. We first sample 100 random architectures from each search space. We then compute all training-free metrics on these architectures using 1000 randomly sampled *clean* data points from CIFAR-10. Then, we recompute the metrics on data points from the *poisoned* datasets and record the change as a percentage. By comparing metric values using the same architecture, we isolate the change caused by the data. We consider poisoning budgets of $p = 50\%$ to assess the robustness in the worst case. Table 2 shows our results.

---

[3]We omit metrics that do not use any data.

We find that the training-free metrics are largely insensitive to data poisoning: differences range from 0–65.42% and remain below 10% in all but two cases. For TE-NAS, $\kappa_{\text{NTK}}$ is the most sensitive, increasing by 65.42% on average by GC, though its high standard deviation indicates instability that may not be a direct consequence of the poisons. For RoBoT, the metrics are less affected, with the largest deviation being -11.14% for *grasp* by CLF. Label-flipping attacks tend to have a larger impact than clean-label, except for *fisher*, which is altered by 2.57% by Gaussian noise. In all, these metric changes are unlikely to significantly alter the algorithms that leverage them, accounting for their comparative robustness to training-based methods.

### 5.3 NAS Generalizes to Out-of-Distribution Data

If NAS generalizes well to out-of-distribution (OOD) datasets, this may explain why poisoning attacks—which effectively introduce OOD data points—are less effective. To quantify this generalization, we run the three NAS algorithms from §3.4 on several OOD training datasets to produce architectures and then train these architectures on CIFAR-10 (the original data). Following our main evaluation, we use ΔImp. to measure the performance and use the same random sampling Acc. of 96.73%. Here, we aim to test whether searching on OOD datasets produces architectures with lower accuracy (compared to searching on CIFAR-10). Accordingly, as in §4.2, we measure significance using one-sided Welch's t-tests at a significance level of $\alpha = 0.05$ (using the BH procedure for FDR control); the null hypothesis is that searching on OOD datasets does not degrade final accuracy. Table 3 presents our results on the MNIST (Deng, 2012), FashionMNIST (Xiao et al., 2017), and SVHN (Netzer et al., 2011) datasets; CIFAR-10 is included as the baseline expected improvement.

The generalization ability of NAS varies substantially across algorithms. TE-NAS generalizes best, with ΔImp. ranging from 0.07–0.22 and no statistically significant drops compared to searching on CIFAR-10. Interestingly, these benefits are equivalent to or exceed running TE-NAS on *in-distribution* training data (CIFAR-10), suggesting that the training distribution need not align with the target task. RoBoT also generalizes well with no statistically significant reductions, although it only improves over CIFAR-10

Table 3: **Out-of-distribution search results.** For each training dataset, we report the percentage-point change in accuracy compared to random sampling (ΔImp.) with the standard deviations. All architectures are re-trained from scratch on CIFAR-10. Statistically significant results are **bolded and underlined**.

| Training Dataset | P-DARTS | | RoBoT | | TE-NAS | |
|---|---|---|---|---|---|---|
| | Acc. | ΔImp. | Acc. | ΔImp. | Acc. | ΔImp. |
| **CIFAR-10** | 97.32 ± 0.07 | 0.58 | 96.99 ± 0.20 | 0.25 | 96.80 ± 0.40 | 0.07 |
| **MNIST** | **97.14** ± 0.12 | **0.41** | 96.94 ± 0.36 | 0.21 | 96.89 ± 0.27 | 0.16 |
| **FashionMNIST** | **97.09** ± 0.21 | **0.36** | 96.88 ± 0.20 | 0.15 | 96.80 ± 0.41 | 0.07 |
| **SVHN** | **96.65** ± 0.13 | **-0.08** | 97.00 ± 0.22 | 0.27 | 96.95 ± 0.19 | 0.22 |

when training on SVHN. P-DARTS performs worst, never exceeding its in-distribution ΔImp., with all three OOD datasets leading to statistically significant degradation as low as -0.08% points below random sampling. Unlike training-free and hybrid NAS, P-DARTS requires in-distribution data to achieve optimal results.

We hypothesize two factors behind this finding: (1) the OOD datasets share low-level features that guide NAS toward general-purpose feature extractors (e.g., convolutional and pooling operations), and (2) training-free and hybrid NAS algorithms tend to generalize well because they evaluate architectures based on representational capacity and trainability rather than performance (Chen et al., 2021; Shu et al., 2022a;b). This makes these algorithms inherently less sensitive to distribution shifts, explaining their robustness to data poisoning attacks and other OOD datasets. However, this robustness comes at a cost: P-DARTS achieves substantially higher ΔImp. than RoBoT and TE-NAS when trained on in-distribution data.

## 6 Discussion

### 6.1 Potential Countermeasures

We now discuss potential countermeasures. Many defenses have been proposed against data poisoning attacks on machine learning. A common approach is *outlier detection* (Paudice et al., 2018; Steinhardt et al., 2017), which leverages the insight that poisoned samples are distinct from clean data. However, knowledgeable attackers (Koh et al., 2021) and clean-label constraints (Geiping et al., 2021) bypass these defenses by crafting indistinguishable poisons from benign data. Other works seek to enhance the robustness of the model through

training, offering either certifiable guarantees on test-time accuracy (Hong et al., 2024; Rosenfeld et al., 2020; Levine & Feizi, 2021) or strong empirical results against existing attacks (Hong et al., 2020; Liu et al., 2022). While effective, certifiable defenses incur significant performance drops, and empirical approaches cannot guarantee robustness against adaptable adversaries. More importantly, it is unclear how countermeasures designed to defend against data poisoning during training transfer to the architecture selection process performed by NAS. Prior defenses focus solely on the final test accuracy of models trained on poisoned data, specifically the quality of the learned weights and biases. In contrast, NAS prioritizes architectural quality.

To assess the applicability of data poisoning defenses on NAS, we evaluate three compatible countermeasures on CIFAR-10. The first, diffusion denoising (Hong et al., 2024), is a certified defense that uses off-the-shelf diffusion models to remove adversarial perturbations. Because it operates before training, its certification is independent of the specific training process, making it suitable for non-standard poisoning scenarios such as NAS. The second is loss-based sanitization (Koh et al., 2021), which removes training examples that are not well-fit by a separate classifier trained on the full dataset. The intuition is that poisons will have higher loss because they differ significantly from clean data points. Third, given our finding that label noise is the most harmful attack vector, we also propose *relabeling* the training data via unsupervised clustering. Specifically, we extract the final-layer features of the CIFAR-10 training set from a pre-trained neural network, cluster them with K-Means (Lloyd, 1982), and assign a common label to each cluster. This process creates entirely new training labels, eliminating adversarial label flips. Although cluster-based relabeling may see limited practicality—since it requires a large-scale pretrained feature extractor—it serves as a valuable conceptual tool for examining whether NAS can better recover from structured rather than adversarial label noise.

Table 4: **Data poisoning countermeasures.** The **ΔImp.** across defenses. All attacks use $p = 50\%$. Statistically significant results are **bolded and underlined.**

| Defense | Attack | P-DARTS | | RoBoT | | TE-NAS | |
|---|---|---|---|---|---|---|---|
| | | **Acc.** | **ΔImp.** | **Acc.** | **ΔImp.** | **Acc.** | **ΔImp.** |
| **Cluster-Based Relabeling** | RLF/CLF | **97.16** ± 0.17 | **0.42** | 96.98 ± 0.18 | 0.24 | – | – |
| **Diffusion Denoising** | Gaussian Noise | 97.29 ± 0.14 | 0.56 | **97.11** ± 0.03 | **0.38** | 96.86 ± 0.29 | 0.12 |
| | GC | 97.20 ± 0.30 | 0.47 | 96.91 ± 0.10 | 0.18 | 96.74 ± 0.34 | 0.01 |
| **Loss-Based Sanitization** | Gaussian Noise | 97.12 ± 0.17 | 0.39 | 97.08 ± 0.09 | 0.34 | 96.72 ± 0.33 | -0.01 |
| | GC | 97.15 ± 0.14 | 0.42 | 97.05 ± 0.00 | 0.32 | 97.04 ± 0.35 | 0.31 |
| | RLF | **97.28** ± 0.14 | **0.55** | **97.15** ± 0.10 | **0.42** | – | – |
| | CLF | 97.11 ± 0.21 | 0.37 | 97.05 ± 0.01 | 0.32 | – | – |

**Results.** We evaluate diffusion denoising against Gaussian noise and Gradient canceling **(GC)** with $p = 50\%$. The denoising parameter $\sigma$ is set to 0.1, as the original study finds it offers the best trade-off between robustness and performance. For loss-based sanitization, we first train a ResNet-18 (He et al., 2016) on each poisoned dataset from §4 (with $p = 50\%$). Following Koh et al. 2021, we then discard the top 50% of training examples in each class with the highest training loss. For our relabeling defense, we extract features with a ResNet-152 model pre-trained on ImageNet (Deng et al., 2009) and group them into 10 clusters. Relabeling discards the original labels, inherently defending against RLF and CLF at any attack budget. To assess effectiveness, we compare the mean accuracy of defended models against poisoned baselines using a one-sided Welch's t-test at a significance level of $\alpha = 0.05$. The null hypothesis assumes that applying the defenses to poisoned datasets does not improve accuracy. Results across ten trials are shown in Table 4.

The defenses show limited effectiveness. Loss-based sanitization performs best, achieving statistically significant improvements in ΔImp. for both P-DARTS and RoBoT against RLF. However, its effectiveness against other attacks drops, indicating less applicability to clean-label or targeted scenarios. Cluster-based relabeling yields a statistically significant improvement in ΔImp. for P-DARTS, but significantly underperforms clean training (ΔImp. of 0.42 vs. 0.58). While it may mitigate label noise, the new labels are likely too inaccurate to restore performance. Diffusion denoising leads to significant improvement only for RoBoT under Gaussian noise (increasing ΔImp. from 0.30 to 0.38), suggesting it is more effective against random perturbations than targeted attacks. However, considering the clean-label attacks were ineffective against RoBoT in §4.2, further investigation is needed to determine the precise impact of this defense.

### 6.2 Future Work

**Expanding our auditing framework.** While we make a significant effort to ensure the representativeness of our framework—investing over 15,000 GPU hours in our main evaluation—evaluating a broader range of NAS algorithms and attacks could further enhance the generalizability of our findings. In particular, incorporating more non-differentiable, training-based algorithms—such as reinforcement learning-based approaches (Zoph & Le, 2017; Baker et al., 2017; Pham et al., 2018)—would allow us to assess additional methods with high data reliance. Furthermore, while our work focuses on the empirical characterization of data poisoning attacks in NAS settings, future research could develop a stronger theoretical foundation, for example, by extending the framework from Lu et al. (2023) to the search phase of NAS. Such a foundation would enable the principled design of NAS-specific data poisoning attacks, potentially overcoming the limitations revealed in our study.

**Improving NAS-specific data poisoning attacks.** While our NAS-specific GC attack induces statistically significant accuracy drops in P-DARTS, it faces several limitations. First, it assumes a white-box setting in which the adversary knows the victim's NAS algorithm—which may not hold in real-world scenarios. Second, it underperforms label-flipping attacks, which we attribute to three factors: (1) NAS pipelines commonly use data augmentations, which hinder clean-label poisoning success (Geiping et al., 2021; Borgnia et al., 2021; Schwarzschild et al., 2021); (2) in many algorithms (including P-DARTS), the supernet evolves during search, which reduces transferability since we craft poisons on a static model; (3) GC targets continuous architectural parameters, but NAS ultimately selects a discretized architecture that may not preserve the targeted "sub-optimality." These limitations reveal several opportunities for improving NAS-specific data poisoning—such as incorporating differentiable data augmentations (Geiping et al., 2021) or regularization to encourage producing near-discretized architectures—which we view as an exciting direction for future work.

**Alternative data poisoning scenarios.** As poisoning during the training phase of neural networks has been extensively studied (Zhao et al., 2025), our auditing framework is specifically designed to evaluate the *novel* component of NAS—the architecture selection process—in isolation. To this end, we poison only the search phase and retrain the resulting architectures on clean data. In practice, however, the same dataset is often used for both search and training, introducing an additional opportunity for poisoning. Simultaneously corrupting the architecture discovery and the subsequent parameter learning could induce greater performance degradation, making these attack vectors more impactful. Moreover, such settings enable more complex attack goals—such as targeted misclassification (Huang et al., 2020; Geiping et al., 2021; Shafahi et al., 2018; Aghakhani et al., 2021) or backdooring (Liu et al., 2018c; Gu et al., 2019)—which require manipulating model parameters rather than the architecture. Because this scenario entails a distinct framework and research scope, we regard it as a valuable direction for future work toward a deeper understanding of NAS's robustness.

## 7 Conclusion

Our work demonstrates that the performance gains achieved by architectures produced through NAS algorithms are *not* robust to data poisoning attacks. To systematically study this vulnerability, we design an auditing framework that enables the execution of various NAS algorithms on datasets compromised by poisoning attacks, including an attack specifically tailored for NAS. Our framework also implements a testing methodology and a specialized metric to effectively evaluate robustness.

In our evaluation, we first find that accuracy alone is insufficient for quantifying vulnerability: it makes NAS algorithms appear robust to data poisoning because their reported performance gains over random search are often marginal. However, our proposed metric, $\Delta$Imp., shows that data poisoning can reduce the benefit of NAS by up to 156%, revealing a much higher level of susceptibility. We also show that NAS algorithms that rely heavily on data are inherently more vulnerable, whereas those with reduced data dependence are more robust but with performance comparable to random sampling. Moreover, our results challenge the validity of the performance improvements achieved by NAS. Surprisingly, we observe that running NAS on completely out-of-distribution datasets—such as MNIST—and then training the produced architectures on CIFAR-10 can yield higher accuracy than running NAS directly on CIFAR-10. These findings raise fundamental questions about the effectiveness and reliability of NAS as a "data-centric" paradigm for neural architecture design.

**Broader Impact Statement**

In recent years, *data-centric* approaches to finding optimal neural network architectures have emerged, namely neural architecture search (NAS). This paper proposes a data poisoning framework for NAS algorithms as an auditing tool for their robustness to underlying data distribution shifts in adversarial settings. While our results suggest that data poisoning only inflicts marginal drops in the accuracy of models produced by these algorithms, this can practically invalidate their effectiveness with respect to the improvement they provide over hand-crafted or even randomly sampled architectures.

Releasing this work and its accompanying framework may pose dual-use risks, as adversaries could potentially exploit our methods to compromise NAS—for example, degrading architectures deployed in reliability-critical domains such as autonomous driving (Liu, 2020) or medical diagnostics (Sarvamangala & Kulkarni, 2022), where even small accuracy losses can have significant consequences. We also do not assess robustness beyond accuracy, where adversaries might instead target other NAS objectives such as latency or model size. Nevertheless, we believe the potential benefits of releasing our framework far outweigh these risks. It enables systematic investigation of NAS vulnerabilities to data poisoning and supports the development of more robust defenses. Furthermore, since the most severe degradations occur only under large poisoning budgets ($p \geq 10\%$), the current real-world risk remains limited, while the expected research benefit is substantial.

Overall, we envision this research to raise awareness regarding the vulnerabilities associated with NAS and its data-centric approach. By demonstrating how an attacker could potentially compromise NAS through the data it uses, we illuminate the importance of verifying training data and promote further research.

**Acknowledgments**

We thank the anonymous reviewers for valuable feedback. We thank Fuxin Li for his feedback on our findings. This work is partially supported by the Google Faculty Research Award 2023. The findings and conclusions in this work are those of the author(s) and do not necessarily represent the views of the funding agency.

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

## A  Data Poisoning Results for Evolutionary NAS

In this section, we evaluate our framework on the evolutionary NAS algorithm NSGANetV2 (Lu et al., 2020). Following Lu et al. (2020), we sample the four best architectures from each search run's Pareto front. Due to resource constraints, we evaluate only the architecture with the third-highest FLOP count (the "large" configuration). Because NSGANetV2 uses a custom search space, we first establish a random-sampling baseline of 97.90% Acc. obtained from 10 randomly sampled architectures. We then apply the poisoning setup and evaluation procedure from §4.2, reporting Acc. and ΔImp. across all attacks and budgets in Table 5.

Table 5: **Robustness of NSGANetV2 to data poisoning.** The **Acc.** and **ΔImp.** across poisoning budgets. Statistically significant results are **bolded and underlined.**

| Poisoning Attack | $p$ | Acc. | ΔImp. |
|---|---|---|---|
| **None** | – | $98.06 \pm 0.19$ | 0.17 |
| **Gaussian Noise** | 1% | $97.97 \pm 0.14$ | 0.08 |
| | 10% | $97.90 \pm 0.18$ | 0.00 |
| | 50% | $97.87 \pm 0.16$ | -0.02 |
| **GC** | 1% | $97.99 \pm 0.19$ | 0.09 |
| | 10% | $98.02 \pm 0.25$ | 0.12 |
| | 50% | $97.85 \pm 0.20$ | -0.04 |
| **RLF** | 1% | $98.02 \pm 0.07$ | 0.12 |
| | 10% | $97.93 \pm 0.21$ | 0.03 |
| | 50% | $97.97 \pm 0.18$ | 0.07 |
| **CLF** | 1% | $98.01 \pm 0.22$ | 0.11 |
| | 10% | $97.98 \pm 0.20$ | 0.09 |
| | 50% | $\underline{\mathbf{97.71}} \pm 0.23$ | **-0.19** |

NSGANetV2 exhibits a statistically significant drop in ΔImp. only under CLF with $p = 50\%$, indicating that it is more robust than the other training-based algorithm, P-DARTS. However, it performs worse on clean data (ΔImp. of 0.17 vs. 0.58), further supporting the utility–robustness trade-off described in §4.2. Overall, the results for NSGANetV2 reinforce our main findings: training-based NAS remains more vulnerable to data poisoning than non-training-based methods (which show no statistically significant performance degradation), and label-flipping attacks outperform clean-label attacks.

## B  Impact of Data Augmentations on NAS-Specific Gradient Canceling

In §4.2, we hypothesize that our NAS-specific gradient canceling (GC) attack may be partially mitigated by data augmentations, as prior work has shown these to be effective countermeasures against clean-label attacks (Geiping et al., 2021; Schwarzschild et al., 2021; Borgnia et al., 2021). To test this hypothesis, we evaluate GC on P-DARTS *without* applying data augmentations to the CIFAR-10 training data during the search phase. For comparison with the accuracies reported in §4.2, the resulting architectures are trained from scratch on clean data *with* data augmentations applied. We assess statistically significant performance drops using the same procedure as in §4.2; significant results are **underlined and bolded**. Table 6 reports the Acc. and ΔImp. for P-DARTS across $p \in \{1, 10, 50\}\%$.

NAS-specific GC is more effective when data augmentation is not applied during the P-DARTS search. Although the same poisoning budgets from §4.2 are statistically significant (1% and 50%), the drop in ΔImp. is up to 2.2× greater. This affirms our hypothesis that data augmentations are an effective countermeasure against targeted clean-label poisoning attacks on NAS.

Table 6: **Effectiveness of NAS-specific GC *without* data augmentations.** The **Acc.** and $\Delta$**Imp.** for P-DARTS across poisoning budgets. Statistically significant results are **bolded and underlined.**

| Poisoning Attack | $p$ | Acc. | $\Delta$Imp. |
|:---:|:---:|:---:|:---:|
| **None** | – | $97.19 \pm 0.15$ | 0.46 |
| **GC** | 1% | $\mathbf{\underline{96.99}} \pm 0.27$ | **0.26** |
| | 10% | $97.11 \pm 0.11$ | 0.38 |
| | 50% | $\mathbf{\underline{96.84}} \pm 0.23$ | **0.11** |

## C  Visualization of Poisoned Images

Here, we provide a visualization of poisons crafted using Gaussian noise (Noise) and Gradient canceling (GC) (Lu et al., 2023). Figure 2 shows several examples; all poisons are crafted with $\ell_\infty$ bounds of $\varepsilon = 16$.

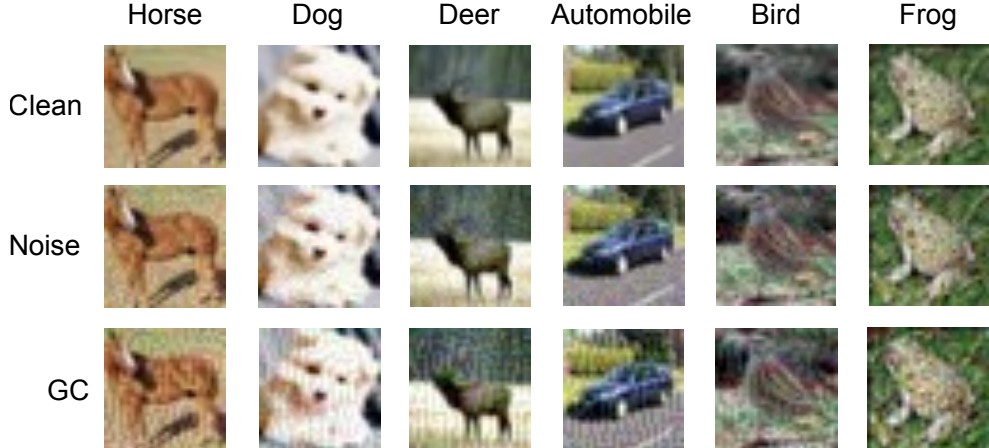

Figure 2: **Visualization of poisons.** A comparison of six randomly drawn images in CIFAR-10. **Clean** refers to clean samples, **GC** is the gradient canceling attack Lu et al. (2023), and **Noise** is Gaussian noise. The other attacks we employ, random and confidence-based label flipping, use clean images but modify labels.

