# OpenReview forum: "Hard Work Does Not Always Pay Off: On the Robustness of NAS to Data Poisoning"
_TMLR — Accepted by TMLR_

### Review · Reviewer_aggA · 2025-09-22

**Summary Of Contributions:**

As neural architecture search (NAS) algorithms are purely data-centric approaches, they could be vulnerable to poisoned data. This issue, however, remains unsolved. To this end, this paper takes the first step to investigate this issue, specifically, “how vulnerable are NAS algorithms to data poisoning attacks?”. They propose a complete pipeline to answer this question, including defining the threat model, designing data poisoning attacks, and evaluating the influence of the attack with newly proposed metrics. Results show that NAS algorithms are somewhat robust to data poisoning attacks.

**Additional Comments:**

1. This paper is the first to explore the vulnerability of NAS algorithms against data poisoning attacks. I think this problem is well-motivated and interesting.

2. The proposed pipeline to explore the vulnerability of NAS algorithms, including defining the threat model, designing data poisoning attacks, and evaluating the influence of the attack with newly proposed metrics, defines a complete framework for future research.

3. It seems the authors consider $\triangle Imp$ using the absolute difference between the accuracy of NAS’s architecture and the accuracy of a random architecture. But shouldn’t we consider a relative difference rather than the absolute difference here? since the accuracy of a random architecture could be very random.

4. My first concern is that, according to the presented results (Table 1), it seems the NAS algorithms themselves are not effective since $\triangle Imp$ is 0.58% at most **when there is no attack**. Given the ineffectiveness of NAS, the results under attacks would be **uninteresting** in the sense that any attack leading to $\triangle  Imp <0.58%$ can be claimed effective, which is not persuasive. (Recall in data poisoning attacks on image classification models, we typically claim an attack is effective when it reduces ACC from about 100% to about 0%).

5. My second concern is that, it seems the attacks are not effective in the sense that the existence of poisoned data may sometimes let NAS find better architectures (as $\triangle  Imp$ is higher), and increasing the poisoning rate does not always worsen NAS performance (which is different from the typical findings in data poisoning attacks on image classification models).

6. Overall, this paper proposes a novel and well-motivated **scenario**—data poisoning attacks on NAS; but, it seems the **performance drop** (i.e., effectiveness of attacks) is not obvious when confronted with the attacks. Nevertheless, the findings could be interesting to related communities.

**Audience:**

Yes

**Audience Explanation:**

NAS algorithms are widely deployed in searching for optimal neural architectures. As these algorithms are purely data-driven, they could be very vulnerable. Thus, the problem investigated by this paper, i.e., the vulnerability of NAS algorithms, is well-motivated and of significance, especially for the NAS and trustworthy AI communities.

**Broader Impact Concerns:**

None.

**Claims And Evidence:**

Yes

**Claims Explanation:**

The claims made in this paper—the vulnerability of NAS to data poisoning attacks—are basically supported with evaluations on the CIFAR-10 dataset and three different types of NAD algorithms using the employed dirty-label and clean-label attacks. Please see more concerns in “Additional Comments”.

**Requested Changes:**

This paper only evaluates the proposed attacks on the CIFAR-10 dataset. It would be appreciated if the authors could increase evaluations on more datasets, which could make the paper's claims more convincing. — I am unfamiliar with NAS, not sure whether CIFAR-10 is typically the only benchmark that this community will use.

---

> ### Author Response · Authors · 2025-10-09
> **Rebuttal by Authors**
>
> We thank the reviewer for the constructive comments on our manuscript. We were able to improve our manuscript substantially based on these comments. Below, we summarize our answers and updates made, and we will be happy to address the reviewer’s comments, if any.
>
> ---
>
> **[Updates on the manuscript]**
>
> > This paper only evaluates the proposed attacks on the CIFAR-10 dataset. It would be appreciated if the authors could increase evaluations on more datasets, which could make the paper's claims more convincing.
>
> We extend our evaluation to the more challenging CIFAR-100 dataset. We include results for all poisoning attacks and NAS algorithms in Table 1 of our manuscript (revisions are in blue). As the GC experiments require a significant amount of compute, we are still working on finalizing the results and will add them to our revision as they are completed.
>
> Overall, our results on CIFAR100 support our central claims: that (1) training-based NAS algorithms are the most vulnerable to data poisoning, (2) label-flipping is a more effective attack vector than clean-label, and (3) the training-free and hybrid approaches are more resilient to data poisoning but significantly underperform the training-based algorithms.
>
> ---
>
> **[Our answers and updates on the additional comments]**
>
> > #1: The authors use the absolute difference between the accuracy of NAS’s architecture and the accuracy of a random architecture. But shouldn’t we consider a relative difference since the accuracy of a random architecture could be very random?
>
> We first clarify that the accuracy of random architectures is not entirely random. On CIFAR-10 and CIFAR-100, their test accuracies are 96.7% and 80.7% with standard deviations of 0.40% and 1.14%, respectively. Importantly, “random” does not mean “all possible architectures”; we exclude candidates with incompatible activation dimensions, as such architectures cannot process data (they result in PyTorch errors) and therefore cannot be trained and used.
>
> $\Delta$Imp. encodes “how much the architecture discovered by NAS improves over the random baseline.” It aligns with prior studies that have focused on pulling this number up by a few percentage points through the design of algorithms that impose empirically effective inductive bias. We prefer $\Delta$Imp. over relative-based metrics as it operates on the native accuracy scale, avoiding any instability caused by large or small accuracy denominators in relative metrics, and it enables consistent comparisons across datasets and NAS algorithms.
>
> ---
>
> > #2: The NAS algorithms themselves are not effective since $ \Delta $Imp. is 0.58% at most when there is no attack. Given the ineffectiveness of NAS, the results under attacks would be uninteresting in the sense that any attack leading to -$\Delta$Imp. can be claimed to be effective, which is not persuasive.
>
> We first clarify that a $\Delta$Imp. of 0.58% *does not* indicate that NAS is ineffective. Improvements of this magnitude are typical in the NAS literature. For example, P-DARTS reports **only a 1% gain over the curated DenseNet-BC [1], yet this result received significant attention**, including an oral presentation at ICCV 2019. Because baseline models (e.g., curated architectures or random sampling) are already strong, improvements are often marginal ($\leq$1%), but still recognized as meaningful contributions in the community.
>
> Second, the objective of data poisoning on NAS is fundamentally different from that on fixed architectures. NAS invests substantial computational resources (beyond training) to obtain architectures that *slightly* outperform hand-crafted ones. If a data poisoning adversary prevents or reverses this gain, the effort and resources devoted to NAS are effectively nullified, completely undermining its practical use. Thus, even though the absolute accuracy drops are smaller than in standard image classification models, the impact on the *viability of NAS itself* is substantial and (we argue) therefore scientifically interesting.
>
> [1] Huang et al. *Densely Connected Convolutional Networks*, CVPR (2017).

---

> ### Author Response · Authors · 2025-10-09
> **Rebuttal by Authors (Cont.)**
>
> > #3: The attacks are not effective as poisoned data may sometimes let NAS find better architectures, and increasing the poisoning rate does not always worsen NAS performance.
>
> We acknowledge that some results may appear unintuitive, such as data poisoning increasing $\Delta$Imp, or smaller budgets outperforming larger ones. However, we do not view these as evidence of attack ineffectiveness, but rather as reflections of the *high variance inherent to NAS*, which has been well-documented in prior work [1, 2]. Random initialization, stochastic search dynamics, and model-training variability all contribute to this noise.
>
> To mitigate this variance, we conducted up to 2.5$\times$ more trials than previous studies and applied statistical testing with FDR control. While these measures substantially reduce noise, it is intractable to eliminate it fully given the scale of our study (note that running our main evaluation on both CIFAR-10 and CIFAR-100 requires ~15,000 GPU hours in total).
>
> Importantly, despite individual anomalies, our results reveal consistent and statistically significant patterns across datasets and NAS algorithms: (1) NAS appears robust to data poisoning, but its relative improvement can be notably diminished; (2) training-based NAS algorithms are substantially more vulnerable than training-free and hybrid approaches; (3) NAS algorithms generalize surprisingly well to OOD datasets, questioning their *data-centric* motivation. These broad and novel takeaways, rather than isolated fluctuations, are the meaningful contributions of our study.
>
> [1] White et al., *Exploring the Loss Landscape in Neural Architecture Search*, Uncertainty in Artificial Intelligence (2021).
>
> [2] Dushatskiy et al., *Heed the Noise in Performance Evaluations in Neural Architecture Search*, GECCO (2022).

---

> > ### Comment · Reviewer_aggA · 2025-10-11
> >
> > Thanks for the authors' responses, which have addressed my concerns.

---

> > > ### Author Response · Authors · 2025-10-11
> > > **Acknowledgment**
> > >
> > > We sincerely thank the reviewer for the constructive feedback and are glad that our clarifications have addressed the raised issues.

---

### Review · Reviewer_9ibG · 2025-09-24

**Summary Of Contributions:**

This paper investigates an important and novel question: how susceptible neural architecture search (NAS) methods are to data poisoning. The authors propose a systematic auditing framework that injects poisons into the data used during the search phase. They implement four poisoning strategies (random label flips, confidence-based label flips, Gaussian noise, and a NAS-specific gradient-canceling attack) and introduce a new metric, ∆Improvement, which measures the drop in performance gain of NAS versus a random baseline. Three representative NAS methods are evaluated: a training-based differentiable NAS (P-DARTS), a training-free NAS (TE-NAS), and a hybrid NAS (RoBoT). Experiments on CIFAR-10 reveal that raw accuracy degrades only marginally under poisoning, but the relative gain from NAS can be greatly diminished (up to 93% loss in ∆Imp). In particular, training-based NAS suffers the most from label-flip poisons, while training-free NAS is largely unaffected (but its architectures offer little benefit above random). A surprising finding is that searching on completely out-of-distribution data (e.g. using MNIST to search architectures for CIFAR-10) can still produce high-performing models. Finally, the authors test two defenses (diffusion denoising and unsupervised relabeling) and find them largely ineffective at restoring the NAS gains.

**Major Strengths:** The paper tackles a gap in the literature by combining two active areas (NAS and data poisoning). The experimental methodology is thorough: it uses repeated trials, proper baselines (random search), and statistical tests to ensure robustness. Introducing the ∆Improvement metric is a clear, justified idea to capture “gain loss” beyond raw accuracy. The results yield non-trivial insights: there is a clear utility–robustness trade-off (high-accuracy NAS are more vulnerable, while low-utility NAS appear robust), and the out-of-distribution search finding is intriguing. The paper is well-organized and the figures/tables effectively illustrate the claims.



**Key Weaknesses:** The scope of evaluation is somewhat narrow. All experiments use CIFAR-10 and a single cell-based search space. It remains unclear if the findings generalize to larger datasets (like ImageNet) or other search spaces. As the authors themselves note, non-differentiable NAS (e.g. evolutionary or RL-based) were not tested. The poisoning threat model is limited: only the search-phase data is poisoned, while final training uses clean data. In practice, NAS often uses the same dataset for search and training; poisoning both could have stronger effects. The newly introduced “NAS-specific” gradient attack requires white-box knowledge and, as shown in Table 1, has a negligible impact compared to simpler label flips. Also, the out-of-distribution result, while interesting, is not fully explained and may be counterintuitive. Overall, while the experiments are extensive, additional analysis or ablations (e.g., on different tasks or search settings) would strengthen the claims.

**Audience:**

Yes

**Audience Explanation:**

The paper lies at the intersection of machine learning security and automated model design, both of which are relevant to TMLR’s readership. The question of whether data-driven NAS methods can be fooled by poisoned data is novel (the authors cite it as an “open question”) and has practical significance for anyone relying on NAS in untrusted environments. The findings have immediate implications: they suggest that a standard NAS pipeline might be sabotaged by tampering with its data, and they expose a surprising phenomenon of out-of-distribution search effectiveness.

**Broader Impact Concerns:**

The authors provide a Broader Impact statement highlighting the importance of data integrity in NAS pipelines. This is appropriate, but it is somewhat one-sided: it warns about vulnerabilities and the need to verify data, yet it does not consider the dual-use aspect. In particular, the paper should acknowledge that the developed poisoning methods (and the insight that NAS is vulnerable) could be exploited by malicious actors to intentionally degrade models (e.g. in shared training dataset scenarios). Similarly, while raising awareness is a positive aim, the authors should ensure that publishing detailed poisoning algorithms does not unduly facilitate attackers. On the other hand, the work could spur beneficial developments (like robust NAS algorithms or data validation tools). Lastly, there is no discussion of impacts beyond model accuracy (e.g. whether poisoned NAS might induce biases or fairness issues); while this may be beyond the scope, a note stating that “we do not evaluate fairness or security beyond accuracy” would clarify the impact boundaries.

**Claims And Evidence:**

Yes

**Claims Explanation:**

The claims are backed by a solid set of experiments and well-chosen baselines. The authors use a proper random-search baseline (accuracy ≈96.73%) and compute ∆Improvement to quantify NAS’s gain. They run 10 trials per condition and employ statistical tests to verify significance, which lends confidence to their observations. The key finding that label-flip poisoning can collapse NAS’s improvement by up to 93% is directly evident in Table 1 and discussed in the text. Likewise, the relative behaviors of P-DARTS, RoBoT, and TE-NAS are consistently supported by data (e.g. Table 1 shows training-free TE-NAS has ∆Imp~0.1 under attacks, versus large drops for P-DARTS). The paper’s methodological argument for the ∆Imp metric is clear and justified. The theoretical content is minimal, but the experimental evidence is complete and transparent. A caveat is that all tests are on CIFAR-10; thus some extrapolation is needed to generalize beyond this dataset.

**Requested Changes:**

- **[Critical] Expand the experimental evaluation.** The current results are limited to CIFAR-10 and one search space. To strengthen the claims, the authors should include at least one additional dataset or domain. For example, an ImageNet (or subset) experiment, or even a different image benchmark (e.g. CIFAR-100), would help verify that the robustness patterns generalize. They should also consider testing a non-differentiable NAS method (e.g. an RL-based or evolutionary algorithm) as suggested in the text. Without these, conclusions about “NAS in general” remain speculative.


- **[Critical] Clarify or analyze the out-of-distribution (OOD) search result.** The paper reports that using MNIST/SVHN to search for CIFAR-10 architectures can yield comparable or even higher ∆Imp. This counterintuitive finding deserves deeper analysis. The authors should provide more discussion (or additional experiments) to explain why OOD search works so well (e.g. is it due to shared feature motifs or simply randomness?). In particular, it would help to include standard deviations or significance levels for the OOD results in Table 3, and to clarify under what conditions the advantage is statistically meaningful.

- **[Critical] Re-examine the “NAS-specific GC” attack.** The tailored gradient-canceling (GC) attack is novel, but the paper shows it has little effect on NAS (Table 1). The authors should either improve its strength or more clearly state its limitations. For example, GC requires detailed knowledge of the NAS process. The authors might explicitly discuss the practicality of this assumption. If GC remains weak, they could consider more targeted attacks (perhaps leveraging the alternative “poison both search and train” scenario mentioned as future work). At minimum, the write-up should highlight why GC underperforms label flips here.


- **[Suggested] Strengthen the defense evaluation.** The two defenses (diffusion denoising and cluster relabeling) are a good start, but currently offer limited insight. The authors should clarify the practicality of cluster-based relabeling (for instance, requiring an ImageNet-pretrained model and enough unlabeled data). It would also be useful to compare with other common defenses (e.g. standard data sanitization or robust training approaches) if feasible. At least, the paper should more explicitly acknowledge that cluster relabeling is somewhat artificial.

---

> ### Author Response · Authors · 2025-10-09
> **Rebuttal by Authors**
>
> We thank the reviewer for the constructive comments on our manuscript. We were able to improve our manuscript substantially based on these comments. Below, we summarize our answers and updates made, and we will be happy to address the reviewer’s comments, if any.
>
> ---
>
> **[Updates on the manuscript]**
>
> > The authors should include at least one additional dataset and a non-differentiable NAS algorithm.
>
> Thank you for these valuable suggestions. We first extend our evaluation to the more challenging CIFAR-100 dataset. We include results for all poisoning attacks and NAS algorithms in Table 1 of our revised manuscript. As the GC experiments require a significant amount of compute, we are still working on finalizing the results and will add them to the final version.
>
> Overall, our results on CIFAR100 support our central claims: that (1) training-based NAS algorithms are the most vulnerable to data poisoning, (2) label-flipping is a more effective attack vector than clean-label, and (3) the training-free and hybrid approaches are more resilient to data poisoning but significantly underperform the training-based algorithms.
>
> Moreover, we evaluate an additional evolutionary NAS algorithm (NSGANetV2 [1]). Our preliminary results show a consistent trend with our main findings. However, due to the limited time for the rebuttals, we were unable to complete the full evaluation. We will finalize and incorporate the complete results in Appendix A of our final version.
>
> [1] Lu et al., *NSGANetV2: Evolutionary Multi-Objective Surrogate-Assisted Neural Architecture Search*, ECCV (2020).
>
> ---
>
> > The authors should provide more discussion (or additional experiments) to explain why OOD search works so well. In particular, it would help to include standard deviations or significance levels for the OOD results in Table 3, and to clarify under what conditions the advantage is statistically meaningful.
>
> *Table 3 reports the standard deviations and significance levels* ($\alpha = 0.05$) for our OOD search results, which we have clarified in the revised manuscript. The accuracy drops for P-DARTS are statistically significant across all OOD datasets, whereas those for TE-NAS and RoBoT are not. OOD search does not generally degrade the performance of these algorithms.
>
> We hypothesize two factors behind this finding: (1) the OOD datasets share low-level features that encourage NAS towards general-purpose feature extractors (e.g., convolutional, pooling operations), and (2) training-free and hybrid NAS algorithms tend to generalize well by evaluating architectures based on representational capacity and trainability rather than dataset-specific performance [1, 2, 3]. As a result, these algorithms are inherently less sensitive to distribution shifts, explaining their robustness to both data poisoning attacks and other OOD datasets. We have included this discussion in the revised manuscript.
>
> [1] Shu et al., *Unifying and Boosting Gradient-Based Training-Free Neural Architecture Search*, NeurIPS (2022).
>
> [2] Chen et al., *Neural Architecture Search on ImageNet in Four GPU Hours: A Theoretically Inspired Perspective*, ICLR (2021).
>
> [3] Shu et al., *NASI: Label- and Data-Agnostic Neural Architecture Search at Initialization*, ICLR (2022).

---

> ### Author Response · Authors · 2025-10-09
> **Rebuttal by Authors (Cont.)**
>
> > The authors should either improve the strength of NAS-specific GC or more clearly state its limitations.
>
> We thank the reviewer for pointing this out. In our revision, we will include a write-up explicitly analyzing why NAS-specific GC underperforms and outline key opportunities for improvement. In particular, we attribute GC’s limited impact to the following factors:
>
> 1. **Data augmentations:**  Modern NAS implementations employ data augmentations, which are known to be effective countermeasures against clean-label data poisoning attacks [1, 2, 3]. We are currently including a new experiment in Appendix C to further study this topic.
>
> 2. **Evolving supernet:** The P-DARTS supernet progressively increases in size and prunes operations over time. Because GC poisons are crafted for a static supernet, this evolution can substantially alter the search dynamics, reducing poison transferability.
>
> 3. **Continuous vs. discretized architectures:** GC perturbs continuous architectural parameters, but NAS ultimately selects a discretized architecture. The targeted “sub-optimal” parameters may therefore not translate into significantly degraded discrete architectures, further limiting the attack’s impact.
>
> In line with our study’s goal—to evaluate the current effectiveness of *existing* attacks—our work takes a first step by revealing key challenges unique to poisoning NAS. We view the development of stronger NAS-specific data poisoning attacks as an exciting direction for future research, for example, by incorporating differentiable data augmentations [1] and regularization that encourages near-discretized architectures. We have included this extended discussion on the limitations and opportunities of NAS-specific data poisoning in Section 6.2 of our revision.
>
> [1] Geiping et al., *Witches' Brew: Industrial Scale Data Poisoning via Gradient Matching*, ICLR (2021).
>
> [2] Schwarzschild et al., *Just How Toxic is Data Poisoning? A Unified Benchmark for Backdoor and Data Poisoning Attacks*, ICML (2021).
>
> [3] Borgnia et al., *Strong Data Augmentation Sanitizes Poisoning and Backdoor Attacks Without an Accuracy Tradeoff*, ICASSP (2021).
>
> ---
>
> > The authors should clarify the practicality of cluster-based relabeling and include other common defenses (e.g., standard data sanitization or robust training approaches).
>
> We clarify that our proposed cluster-based relabeling defense is primarily intended as a conceptual tool to assess whether NAS can maintain performance under structured label noise induced by feature-space clustering. This approach is valuable because it inherently mitigates label-flipping attacks by construction, offering insight into the robustness of NAS under controlled versus adversarial perturbations. We acknowledge, however, that its practicality may be limited in real-world settings (e.g., requiring a large-scale pretrained feature extractor such as ResNet-152 trained on ImageNet). We have clarified this point in the revised manuscript.
>
> To further strengthen our defense evaluation, we have incorporated the loss-based data sanitization defense proposed by Koh et al. [1]. These experiments are currently in progress, and the results will be included in Table 4, with additional discussion in Sec 6.1 of the revised manuscript.
>
> [1] Koh et al., *Stronger Data Poisoning Attacks Break Data Sanitization Defenses*, Machine Learning (2022).
>
> ---
>
> **[Updates on the broader impacts]**
>
> We thank the reviewer for this valuable feedback. In response, we have substantially revised the Broader Impact section to contextualize our work better and address the concerns raised:
>
> 1. **Dual-use risks:** We now include an explicit discussion of the dual-use implications associated with developing and releasing NAS-specific poisoning attacks.
>
> 2. **Adversarial consequences:** We expanded the discussion to highlight potential consequences of adversarial NAS manipulation, such as the deployment of lower-quality architectures in high-stakes environments.
>
> 3. **Scope of adversarial objectives**: We clarify that the adversarial objective we consider is performance (accuracy) degradation; future work can explore attacks targeting other objectives, such as latency or model size.
>
> 4. **Scope of claims:** We clarify that severe accuracy degradation (near zero or negative $\Delta$Imp.) is only achieved in our stress-testing scenarios ($p \geq 10$%), suggesting that worst-case outcomes are possible but not currently feasible.
>
> If the reviewer has additional questions or feedback regarding our broader impacts, we are happy to address them.

---

> ### Author Response · Authors · 2025-10-09
> **Rebuttal by Authors (Cont.)**
>
> **[Our answers and updates on the additional comments]**
>
> > The poisoning threat model is limited: only the search-phase data is poisoned, while final training uses clean data.
>
> We clarify that the scope of our work is auditing the novel component of NAS—the architecture selection process—under data distribution shifts induced by poisoning. Considering that data poisoning during the training phase has been extensively studied [1], we find it more impactful to first design and evaluate a framework specifically tailored to this unique aspect of NAS. To this end, our framework is deliberately designed to isolate the causal effect of poisoned data on architecture selection; including poisoning during final training would conflate the effects of search and training, obscuring the mechanisms we aim to study.
>
> We agree that poisoning both phases would capture a broader range of threat models. However, this would require a distinct framework and research focus, which we view as a valuable direction for future work. We have revised our manuscript to clarify this positioning.
>
> [1] Zhao et al., *Data Poisoning in Deep Learning: A Survey*, arXiv preprint (2025).

---

### Review · Reviewer_suW9 · 2025-09-26

**Summary Of Contributions:**

The paper investigates the robustness of neural architecture search (NAS) methods against data poisoning attacks, a relatively underexplored topic. The authors introduce an auditing framework that consists of four poisoning strategies, including a novel adaptation of gradient canceling specifically for NAS, and evaluate their impact across three representative NAS methods: training-based, training-free, and hybrid approaches. A key methodological contribution is the use of a new metric, delta (of improvement), which measures the performance gain of NAS over random sampling, enabling a more precise assessment of how poisoning undermines the marginal improvements NAS typically achieves. Through systematic experiments on CIFAR-10, the findings reveal that while the accuracy drops under poisoning are small, the actual benefits of NAS can be diminished by up to 93%, with training-based methods proving the most vulnerable and training-free methods the most resilient but least useful. The findings also highlight a utility-robustness tradeoff, as methods that rely more heavily on data achieve greater gains but are more susceptible to poisoning, and show that NAS surprisingly generalizes even when searching on out-of-distribution datasets such as MNIST.


The strengths of the paper lie in its clear framing of an overlooked issue, the development of a principled evaluation framework, and the empirical work on running multiple attacks across different NAS paradigms with statistical testing. The introduction of delta improvement is a useful step toward better evaluation of NAS vulnerabilities beyond raw accuracy. However, the work also has notable weaknesses. The study is limited to a narrow scope of methods and datasets (CIFAR-10 within the DARTS search space), raising questions about generalizability. The proposed NAS-specific attack is only a minor modification of existing methods, and most attacks prove relatively ineffective in absolute terms. The defense analysis is shallow, with only two countermeasures tested, both showing limited success. Finally, the reliance on marginal NAS improvements (fractions of a percentage point) as the core signal of vulnerability risks appears to be overstating the practical impact of the findings. Overall, the paper makes a useful first step in connecting NAS and data poisoning but is constrained by methodological and evaluative limitations.

**Additional Comments:**

**Final Note:** Overall, this submission addresses an important but underexplored question of how NAS behaves under data poisoning. The paper is timely, given the increasing reliance on automated model design, and the auditing perspective is worthwhile. That being said, the current work has some limitations that reduce its impact. The experimental scope is narrow (CIFAR-10, a single search space, and only three NAS algorithms), making it difficult to generalize the conclusions. The proposed NAS-specific attack is relatively weak, and the evidence for practical risk is limited, as most attacks yield negligible accuracy degradation. The introduction of the delta of Improvement is interesting, but at times it appears to be used mainly to inflate the perceived severity of the results, since relative reductions are emphasized even when absolute changes are tiny. The writing also tends to overstate findings, for example, claiming NAS is unreliable, without adequate empirical backing. Addressing these issues would significantly strengthen the contribution, both in rigor and in practical relevance.

**Audience:**

Yes

**Audience Explanation:**

Although NAS has matured in recent years, with much work shifting toward training-free, hybrid, or hardware-aware methods, its security and reliability under adversarial conditions remain underexplored. This paper touches on that gap by examining how data poisoning can undermine the marginal gains typically reported by NAS. Even though the evaluation is limited in scope and the practical impact is debatable, some individuals in TMLR’s audience, especially those working on NAS, AutoML, or robustness, would find it useful to see empirical evidence that NAS’s improvements may be fragile under adversarial data. The findings may not change practice immediately, but they highlight potential vulnerabilities and open questions about the long-term reliability of NAS as a data-centric paradigm.

**Broader Impact Concerns:**

The paper includes a Broader Impact Statement, but key issues are underdeveloped. While framed as an auditing study, it introduces NAS-specific poisoning attacks without addressing the dual-use risks this creates. The claims about NAS unreliability are overstated given that the observed accuracy drops are marginal and the evaluation is restricted to CIFAR-10 and a single search space. The discussion should more explicitly acknowledge dual-use implications, avoid overgeneralization, and consider the consequences of adversarial NAS manipulation in high-stakes domains.

**Claims And Evidence:**

No

**Claims Explanation:**

The evidence provided in the paper is limited in scope and does not convincingly support the broad claims that NAS is fundamentally unreliable under data poisoning. The experiments are confined to CIFAR-10 and the DARTS search space, which are saturated benchmarks where random architectures already achieve very high accuracy. As a result, the observed performance drops are marginal in absolute terms, even though the proposed delta Improvement metric appears to inflate their relative impact. The new poisoning attack tailored for NAS is only a small modification of existing methods and proves largely ineffective, with most attacks failing to produce meaningful degradation. Furthermore, only three NAS algorithms are tested, excluding major families such as evolutionary or reinforcement learning–based NAS, which undermines the generality of the findings. The defense analysis is similarly small, evaluating only two countermeasures with predictable outcomes. Overall, while the paper identifies an important question, the narrow setup, reliance on marginal effects, and lack of compelling results mean the claims need to be more convincing or clearly substantiated.

**Requested Changes:**

1. The evaluation could be broadened to include larger and more challenging datasets such as CIFAR-100, Tiny-ImageNet, or an ImageNet subset, since relying solely on CIFAR-10 may overstate or understate the practical impact of poisoning.


2. It would strengthen the paper to incorporate additional NAS algorithms, particularly evolutionary and reinforcement learning–based methods, so that the conclusions better reflect the broader NAS landscape.


3. Extending the analysis across multiple search spaces (e.g., NAS-Bench-201 or latency-aware mobile spaces) rather than restricting to the DARTS space would help ensure that the findings are not space-specific.


4. The study could place more emphasis on realistic poisoning budgets in the 0.1–1% range and treat extreme settings like 50% poisoning as stress-test scenarios rather than representative cases.


5. Evaluating poisoning during both the search and training phases would make the threat model more realistic, since in practice the same dataset often underlies both.


6. The NAS-specific gradient canceling attack would benefit from a clearer theoretical foundation and a principled comparison against stronger baselines such as gradient matching.


7. It would be useful to re-run clean-label attacks with and without data augmentations to better quantify how augmentation choices blunt or amplify attack effectiveness.


8. The ΔImprovement metric should be more carefully justified, ideally by reporting both absolute accuracy changes and ΔImp., and by comparing against other baselines such as top-k random search.

---

> ### Author Response · Authors · 2025-10-09
> **Rebuttal by Authors**
>
> We thank the reviewer for the constructive comments on our manuscript. We were able to improve our manuscript substantially based on these comments. Below, we summarize our answers and updates made, and we will be happy to address the reviewer’s comments, if any.
>
> ---
>
> **[Updates on the manuscript]**
>
> > #1: The evaluation could be broadened to include larger and more challenging datasets such as CIFAR-100, Tiny-ImageNet, or an ImageNet subset, since relying solely on CIFAR-10 may overstate or understate the practical impact of poisoning.
>
> We extend our evaluation to the more challenging CIFAR-100 dataset. We include results for all poisoning attacks and NAS algorithms in Table 1 of our manuscript (revisions are in blue). As the GC experiments require a significant amount of compute, we are still working on finalizing the results and will add them to our revision as they are completed.
>
> Overall, our results on CIFAR100 support our central claims that (1) training-based NAS algorithms are the most vulnerable to data poisoning, (2) label-flipping is a more effective attack vector than clean-label, and (3) the training-free and hybrid approaches are more resilient to data poisoning but significantly underperform the training-based algorithms.
>
> ---
>
> > #2: It would strengthen the paper to incorporate additional NAS algorithms, particularly evolutionary and reinforcement learning–based methods, so that the conclusions better reflect the broader NAS landscape.
>
> We thank the reviewer for this valuable suggestion. We evaluate an additional evolutionary NAS algorithm (NSGANetV2 [1]). Our preliminary results show a consistent trend with our main findings. However, due to the limited time for the rebuttals, we were unable to complete the full evaluation. We will finalize and incorporate the complete results in Appendix A of our final version.
>
> [1] Lu et al., “NSGANetV2: Evolutionary Multi-Objective Surrogate-Assisted Neural Architecture Search”, ECCV (2020).
>
> ---
>
> > #3: Extending the analysis across multiple search spaces (e.g., NAS-Bench-201 or latency-aware mobile spaces) rather than restricting to the DARTS space would help ensure that the findings are not space-specific.
> We first clarify that precomputed tabular benchmarks (e.g., NAS-Bench-101 [1], NAS-Bench-201 [2]) cannot be used in our setting, as our data-poisoning setup requires searching on modified datasets. While extending our analysis to additional search spaces could further test generality, such experiments are computationally prohibitive and beyond the resources typically available to an academic institution. Note that running our main evaluation on both CIFAR-10 and CIFAR-100 requires ~15,000 GPU hours in total.
>
> We also note that, to sustain the generality of our results, we focus on NAS algorithms that represent the most recent and widely adopted trends: supernet-based and differentiable approaches. These algorithms typically employ cell-based search spaces. We adopt the DARTS search space because it minimizes the likelihood that our results are specific to a particular space. It is widely used in the literature, ensuring compatibility with open-source NAS frameworks and facilitating comparison with prior work. Moreover, it is substantially more expressive—spanning up to $10^{13}$ more candidate architectures than popular alternatives [1, 2]—and supports a richer set of operations and graph topologies [3].
>
> [1] Ying et al., *NAS-Bench-101: Towards Reproducible Neural Architecture Search*, ICML (2019).
>
> [2] Dong & Yang, *NAS-Bench-201: Extending the Scope of Reproducible Neural Architecture Search*, ICLR (2020).
>
> [3] Renbo et al., *NAS-Bench-360: Benchmarking Neural Architecture Search on Diverse Tasks*, NeurIPS (2022).
>
> ---
>
> > #4: The study could place more emphasis on realistic poisoning budgets in the 0.1–1% range and treat extreme settings like 50% poisoning as stress-test scenarios rather than representative cases.
>
> Thanks for this suggestion; we agree that it better contextualizes our results. We have revised our manuscript to emphasize the realistic poisoning budget (1%), while treating our other budgets (10% and 50%) as stress-testing scenarios that reveal worst-case outcomes.

---

> ### Author Response · Authors · 2025-10-09
> **Rebuttal by Authors (Cont.)**
>
> > #5: Evaluating poisoning during both the search and training phases would make the threat model more realistic, since in practice the same dataset often underlies both.
>
> We clarify that the scope of our work is auditing the novel component of NAS—the architecture selection process—under data distribution shifts induced by poisoning. Considering that data poisoning during the training phase has been extensively studied [1], we find it more impactful to first design and evaluate a framework specifically tailored to this unique aspect of NAS. To this end, our framework is deliberately designed to isolate the causal effect of poisoned data on architecture selection; including poisoning during final training would conflate the effects of search and training, obscuring the mechanisms we aim to study.
>
> We agree that poisoning both phases would more closely resemble real-world scenarios. But this would require a distinct framework and research focus, which we view as a valuable direction for future work. We have revised our manuscript to further clarify this positioning.
>
> [1] Zhao et al., *Data Poisoning in Deep Learning: A Survey*, arXiv preprint (2025).
>
> ---
>
> > #6: The NAS-specific gradient canceling attack would benefit from a clearer theoretical foundation and a principled comparison against stronger baselines such as gradient matching.
>
> We thank the reviewer for this insightful comment. Our work focuses on the empirical characterization of data poisoning attacks in NAS settings rather than establishing a new theoretical framework. We agree that a stronger theoretical foundation would further strengthen our contribution. While a theoretical analysis is beyond the current manuscript’s empirical scope, we will include a discussion of its connection and leave it as future work in the final version.
>
> We note that Gradient Matching (GM) is excluded from our comparison because it underperforms GC in the *indiscriminate attack* setting. GM aligns the gradients of poisoned data with the target gradients, whereas GC accounts for magnitude by completely offsetting them, as also demonstrated in the original GC paper [1].
>
> [1] Lu et al., *Exploring the Limits of Model-Targeted Indiscriminate Data Poisoning Attacks*, ICML (2023).
>
> ---
>
> > #7: It would be useful to re-run clean-label attacks with and without data augmentations to better quantify how augmentation choices blunt or amplify attack effectiveness.
>
> We thank the reviewer for pointing this out. Our analysis incorporates data augmentation to better reflect realistic NAS pipelines and avoid overestimating the vulnerability. Prior work has shown that data augmentation can reduce attack success [1]. We agree that evaluating the setting without data augmentation would serve as a valuable ablation study. Due to the limited computing budget during the rebuttal period, we will conduct these additional analyses during the revision process and include them in the final version. Please refer to Appendix C for a placeholder.
>
> [1] Schwarzschild et al., *Just How Toxic is Data Poisoning? A Unified Benchmark for Backdoor and Data Poisoning Attacks*, ICML (2021).
>
>
> ---
>
> > #8: The $\Delta$Imp. metric should be more carefully justified, ideally by reporting both absolute accuracy changes and $\Delta$Imp., and by comparing against other baselines such as top-k random search.
>
> The $\Delta$Imp. metric is designed to capture the *relative* change in performance before and after poisoning, enabling fair comparison across NAS algorithms with different baseline accuracies. To make it more interpretable, we will include the absolute accuracy (and its changes) in our final version.
>
> Because top-k random search is a search strategy rather than a metric, we interpret the reviewer’s suggestion as comparing against the top-k architectures obtained from random search. However, we believe this would result in an unfair comparison, as random search explicitly selects the “k” best-performing architectures, whereas other NAS algorithms lack access to such oracle information. If every NAS algorithm were allowed to select its top-k architectures over multiple runs, the setup would effectively reduce to our current evaluation protocol.

---

> ### Author Response · Authors · 2025-10-09
> **Rebuttal by Authors (Cont.)**
>
> **[Updates on the broader impacts]**
>
> We thank the reviewer for this valuable feedback. In response, we have revised the Broader Impact section to better contextualize our work and address the concerns raised:
>
> 1. **Dual-use risks:** We now include an explicit discussion of the dual-use implications associated with developing and releasing NAS-specific poisoning attacks.
>
> 2. **Adversarial consequences:** We expanded the discussion to highlight potential consequences of adversarial NAS manipulation, such as the deployment of lower-quality architectures in high-stakes environments.
>
> 3. **Scope of adversarial objectives**: We clarify that the adversarial objective we consider is performance (accuracy) degradation; future work can explore attacks targeting other objectives, such as latency or model size.
>
> 4. **Scope of claims:** We clarify that severe accuracy degradation (near zero or negative $\Delta$Imp.) is only achieved in our stress-testing scenarios ($p \geq 10$%), suggesting that worst-case outcomes are possible but not currently feasible.
>
> If the reviewer has additional questions or feedback regarding our broader impacts, we are happy to address them.
>
> ---
>
> **[Our answers and updates on the additional comments]**
>
> > The evidence for practical risk is limited, as most attacks yield negligible accuracy degradation.
>
> We argue that, in the context of NAS, even marginal accuracy degradation is evidence of practical risk. The rationale of NAS is to trade extra computational resources (beyond training) for architectures that *slightly* improve over hand-crafted alternatives. For instance, P-DARTS achieved a modest accuracy gain of only $\sim$1% over the curated DenseNet-BC [1], yet this improvement received significant attention from the community, including an oral presentation at ICCV 2019. If a data-poisoning adversary can prevent or reverse such gains, the investment in NAS is effectively wasted, thereby undermining its practical utility. Thus, even small accuracy drops threaten practical viability.
>
> [1] Huang et al. *Densely Connected Convolutional Networks*, CVPR (2017).
>
> ---
>
> > The introduction of $\Delta$Imp. is interesting, but at times it appears to be used mainly to inflate the perceived severity of the results, since relative reductions are emphasized even when absolute changes are tiny.
>
> Following our previous answer, NAS is designed to achieve a tiny absolute improvement over curated architectures. Therefore, we believe that evaluating the relative impact of data poisoning through our $\Delta$Imp. metric is vital for properly contextualizing our results.
>
> ---
>
> > The writing also tends to overstate findings, for example, claiming NAS is unreliable, without adequate empirical backing.
>
> We thank the reviewer for this valuable feedback. We will revise the manuscript to temper any overstatements and ensure that the empirical evidence fully supports all claims. If the reviewer has specific sections in mind where the discussion appears overstated, we would be happy to address them directly.
>
> ---
>
> **[Final remarks]**
>
> We thank the reviewer for their valuable comments, which have substantially strengthened our paper. To summarize, we have added (or are in the process of adding) the following experiments:
>
> 1. Data poisoning results for the CIFAR-100 dataset (Table 1).
>
> 2. Data poisoning results for an additional NAS algorithm, NSGANetV2 (to be added to Appendix A).
>
> 3. Evaluating NAS-specific GC poisoning on P-DARTS without data augmentations (to be added to Appendix C).
>
> 4. A sanitization data poisoning defense (to be added to Table 4).
>
> We believe these additions sufficiently reinforce the empirical evidence supporting our claims, resulting in a more thorough and convincing contribution to trustworthy ML.
>
> We are happy to address any remaining questions or concerns during the rebuttal period.

---

> > ### Comment · Reviewer_suW9 · 2025-10-26
> > **Respond to authors**
> >
> > I thank the authors for their detailed and thoughtful rebuttal, as well as for the substantial revisions made in response to the review. I appreciate the inclusion of new experiments on CIFAR-100 and the preliminary evaluation of an additional NAS algorithm (NSGANetV2), both of which address concerns about generality. The clarifications regarding the infeasibility of using tabular benchmarks in a poisoned-data setting and the justification for focusing on the DARTS search space are reasonable and well-explained.
> >
> > The discussion around realistic poisoning budgets, clearer positioning of the threat model (search vs. training phase), and expanded treatment of broader impact and dual-use risks substantially improve the framing of the work. The authors’ plan to include ablation results on data augmentation and additional GC experiments in the final version is also appreciated.
> >
> > While some aspects, such as the theoretical grounding of the NAS-specific attack and broader search-space coverage, remain open for future work, the revisions meaningfully strengthen the scope and interpretability. Overall, I find the authors’ responses satisfactory and appreciate the improvements made to the paper.

---

### Decision · Action_Editor_tk5G · 2025-11-08

**Recommendation:** Accept with minor revision

**Additional Comments:**

NSGANetV2 results should be updated in the final revision.

**Audience:**

Yes

**Audience Explanation:**

Neural architecture search (NAS) and data poisoning attacks are important research topics in the AutoML and security communities, respectively. Since this paper lies at the intersection of these two areas, I agree with the reviewers that it would be of interest to the TMLR community.

**Claims And Evidence:**

Yes

**Claims Explanation:**

As all reviewers noted, the claims made in this paper are supported by well-designed experiments covering various types of NAS algorithms (training-based, training-free, and hybrid) and multiple poisoning attack settings (both dirty-label and clean-label attacks).

---

> ### Author Response · Authors · 2025-12-09
> **Camera-Ready Revision Update**
>
> Dear Action Editor tk5G,
>
> We have uploaded an updated version of our camera-ready paper addressing the required revisions. Please let us know if any additional changes are needed.
>
> Thank you for your time and consideration.
>
> Warm regards,
> Authors of Paper 5797